# Histone variants shape chromatin states in Arabidopsis

**Bhagyshree Jamge**[1,2†], **Zdravko J Lorković**[1†], **Elin Axelsson**[1†], **Akihisa Osakabe**[1,3,4], **Vikas Shukla**[1,2], **Ramesh Yelagandula**[1,5‡], **Svetlana Akimcheva**[1], **Annika Luisa Kuehn**[1], **Frédéric Berger**[1*]

[1]Gregor Mendel Institute, Austrian Academy of Sciences, Vienna BioCenter, Vienna, Austria; [2]Vienna BioCenter, Vienna, Austria; [3]Department of Biological Sciences, Graduate School of Science, The University of Tokyo, Hongo, Bunkyo-ku, Tokyo, Japan; [4]PRESTO, Japan Science and Technology Agency, Honcho, Kawaguchi, Japan; [5]Institute of Molecular Biotechnology, IMBA, Dr. Bohr-Gasse 3, Vienna, Austria

**\*For correspondence:**
Frederic.berger@gmi.oeaw.ac.at

†These authors contributed equally to this work

**Present address:** ‡Laboratory of Epigenetics, Cell fate & Disease, Centre for DNA Fingerprinting and Diagnostics (CDFD), Hyderabad, India

**Competing interest:** The authors declare that no competing interests exist.

**Summary** How different intrinsic sequence variations and regulatory modifications of histones combine in nucleosomes remain unclear. To test the importance of histone variants in the organization of chromatin we investigated how histone variants and histone modifications assemble in the *Arabidopsis thaliana* genome. We showed that a limited number of chromatin states divide euchromatin and heterochromatin into several subdomains. We found that histone variants are as significant as histone modifications in determining the composition of chromatin states. Particularly strong associations were observed between H2A variants and specific combinations of histone modifications. To study the role of H2A variants in organizing chromatin states we determined the role of the chromatin remodeler DECREASED IN DNA METHYLATION (DDM1) in the organization of chromatin states. We showed that the loss of DDM1 prevented the exchange of the histone variant H2A.Z to H2A.W in constitutive heterochromatin, resulting in significant effects on the definition and distribution of chromatin states in and outside of constitutive heterochromatin. We thus propose that dynamic exchanges of histone variants control the organization of histone modifications into chromatin states, acting as molecular landmarks.

## eLife assessment

This study presents an **important** description on the dynamics of histone variant exchange controlling the organization of the chromatin state of the Arabidopsis genome, combining the analysis of histone variants, histone modification, and chromatin states. The evidence supporting the claims of the authors is **compelling**. This work will be of great interest to those in the field of epigenetics and chromatin biology.

## Introduction

Eukaryotic genomes are packaged into chromatin, a structure defined by repeating units of ~147 bp of DNA wrapped around a protein complex known as the nucleosome (*Luger et al., 1997*). Nucleosomes contain two copies of each of the core histones H2A, H2B, H3, and H4. Histone variants have arisen through the divergence of their intrinsically disordered loop and tail regions and dozens of histone variants in the H2A, H2B, and H3 families have been identified in eukaryotes (*Loppin and Berger, 2020*; *Talbert and Henikoff, 2021*). These variants regulate the properties of nucleosomes (*Chakravarthy et al., 2004*; *Horikoshi et al., 2013*; *Koyama and Kurumizaka, 2018*; *Tachiwana et al., 2012*) and affect transcription (*Rudnizky et al., 2016*; *Subramanian et al., 2015*; *Weber and*

*Henikoff, 2014*; *Wollmann et al., 2012*; *Wollmann et al., 2017*). Animals and plants have evolved unique H2A variants associated with transcriptional repression (*Loppin and Berger, 2020*; *Talbert and Henikoff, 2021*) and with gametes (*Osakabe and Molaro, 2023*). In vascular plants, the H2A.W variants are distinguished by a C-terminal KSPK motif (*Lei et al., 2021*; *Bourguet et al., 2021*; *Yelagandula et al., 2014*). In synergy with H3K9 methylation, H2A.W favors heterochromatic silencing by directly altering the interaction between H2A.W and DNA (*Bourguet et al., 2021*; *Osakabe et al., 2021*; *Bourguet et al., 2021*; *Yelagandula et al., 2014*). The roles of the recently recognized H2B variants (*Jiang et al., 2020*; *Raman et al., 2022*) in transcription remains unknown. However, H2B variants might differ in their capacity to be ubiquitinated, an important modification for RNA polymerase II elongation (*Feng and Shen, 2014*; *Kim et al., 2009*; *Minsky et al., 2008*). Histone variants thus represent complex, diverse nucleosome components that could regulate and organize the functional activities of chromatin.

Histones are subject to a wide range of post-translational modifications (*Talbert and Henikoff, 2021*; *Tessarz and Kouzarides, 2014*). Applying a Hidden Markov Model (ChromHMM) (*Ernst and Kellis, 2012*; *Ernst and Kellis, 2017*) to genomic profiles of histone modifications revealed chromatin states that reflect the combination of post-transcriptional histone modifications present in mammals, *Drosophila*, *Arabidopsis*, and rice (*Ernst and Kellis, 2012*; *Ernst and Kellis, 2017*; *Kharchenko et al., 2011*; *Liu et al., 2018*; *Roudier et al., 2011*; *Sequeira-Mendes et al., 2014*). Chromatin states distinguish the major domains of chromatin within the genome, comprising euchromatin (with active genes), facultative heterochromatin (with repressed genes), and constitutive heterochromatin (with transposons and repeats) in animals and plants. They also highlight specific features such as promoters and enhancers.

Compared with the wealth of data on the relationship between histone modifications and transcription, the role of histone variants in this process remains to be explored. In *Arabidopsis*, there is a clear correlation between transcript levels and enrichment of the variant H3.3 (*Stroud et al., 2012*; *Wollmann et al., 2012*; *Wollmann et al., 2017*) and H2A and H2A.X (*Yelagandula et al., 2014*) on gene bodies marked by H3K36me2 (*Leng et al., 2020*). In contrast, H2A.Z is enriched on repressed genes marked by H3K27me3 (*Carter et al., 2018*), and the histone variant H2A.W is present on transposons and repeats marked by H3K9me2 (*Osakabe et al., 2021*; *Yelagandula et al., 2014*). Although it was suggested that histone variants might have a global role in defining chromatin states (*Hake and Allis, 2006*; *Henikoff et al., 2004*; *Weber and Henikoff, 2014*) the complex, intimate associations between histone variants and modifications have not been fully characterized in plants or mammals.

Here, we analyzed how all thirteen histone variants expressed in vegetative tissues associate with twelve prominent histone modifications to form chromatin states in the model flowering plant *Arabidopsis thaliana*. Our findings indicate that H2A variants are major factors that differentiate euchromatin, facultative heterochromatin, and constitutive heterochromatin. This hypothesis is supported by in silico analyses and the mis-assembly of chromatin states caused by the deregulation of the exchange of H2A variants.

## Results

### Co-occurrence of H3 modifications and histone variants in nucleosomes

In *Arabidopsis*, homotypic nucleosomes containing a single type of H2A variant are prevalent (*Osakabe et al., 2018*). However, it was not determined whether H3 variants also assemble in a homotypic manner and if H2A variants preferably assemble with a specific H3 variant or histone modification. To answer these questions, we immunoprecipitated mononucleosomes (*Figure 1—figure supplement 1A-C*) from transgenic *Arabidopsis* lines expressing HA-tagged *HTR13* (H3.1) and *HTR5* (H3.3) under the control of their native promoters in the respective mutant backgrounds (*Jiang and Berger, 2017*). Because tagged H3 ran slower than endogenous H3 on SDS-PAGE (*Figure 1—figure supplement 1B*), this approach allowed us to analyze the composition of H3.1 and H3.3 nucleosomes by mass spectrometry (MS) and immunoblotting. Spectral counts of peptides covering regions around lysine 27 (K27), where H3.1 and H3.3 can be distinguished by MS analysis, revealed that ~60% of H3.1 and ~42% of H3.3 nucleosomes contained both H3 variants, (*Figure 1A*). We analyzed H3.1 and H3.3 nucleosomes for the presence of H2A variants by western blotting and found that neither H3.1 nor H3.3 was preferentially associated with a specific H2A variant (*Figure 1B*). This was further confirmed

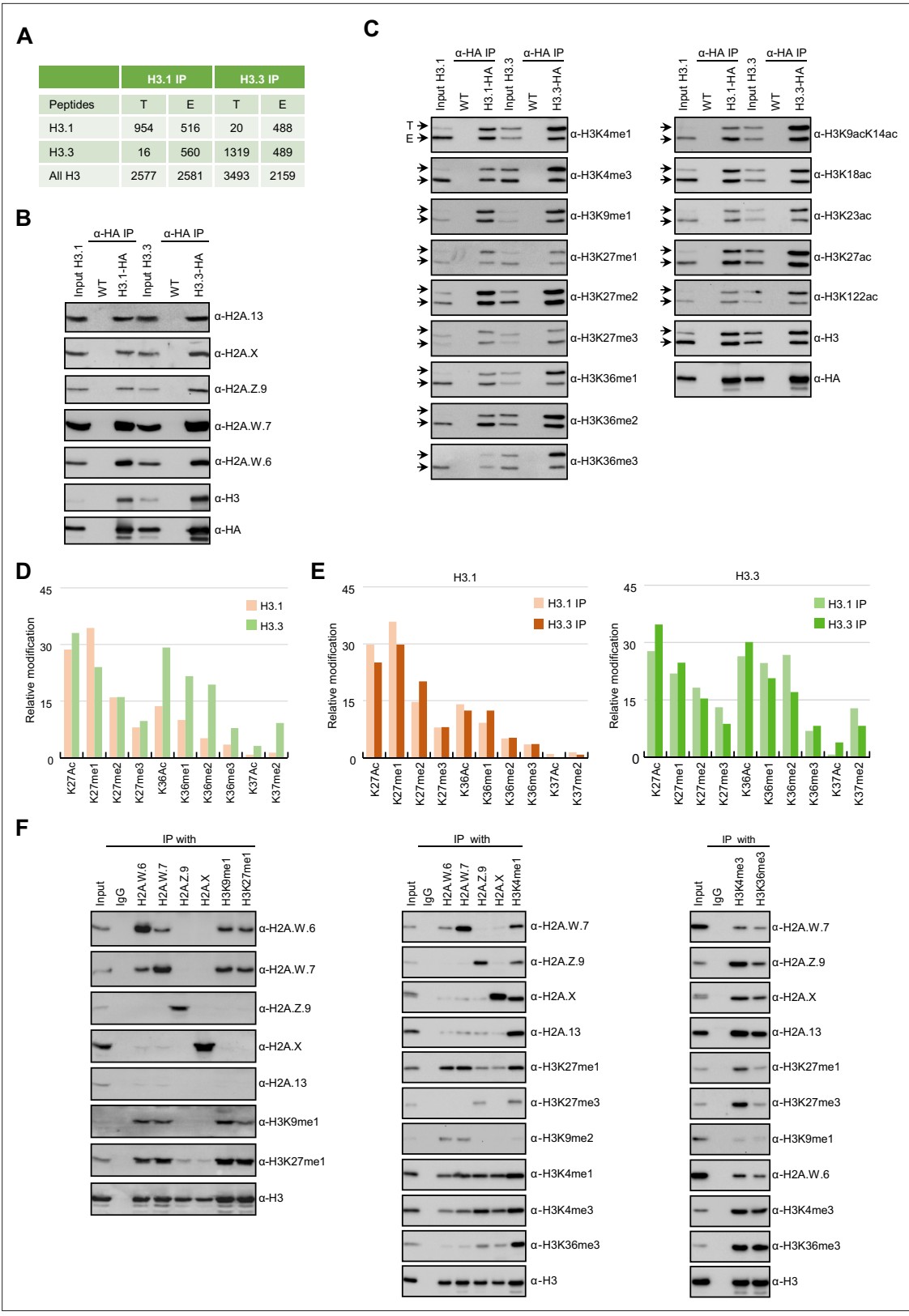

**Figure 1.** Biochemical analysis of the association between histone variants and histone marks. (**A**) Histones H3.1 and H3.3 form homotypic and heterotypic nucleosomes. Spectral counts of H3.1- and H3.3-specific peptides in the respective immunoprecipitations (T – transgenic, E – endogenous H3.1 and H3.3). (**B**) H2A variants do not preferentially associate with H3.1- or H3.3-containing nucleosomes. HA-tagged H3.1 and H3.3 mononucleosomes were immunoprecipitated with HA agarose and analyzed for the presence of H2A variants by immunoblotting. (**C**) Histone H3 marks

*Figure 1 continued on next page*

*Figure 1 continued*

are present on both H3.1 and H3.3. HA-tagged H3.1 and H3.3 mononucleosomes were immunoprecipitated with HA agarose and analyzed for the presence of H3 marks by immunoblotting. Arrows indicate transgenic (**T**) and endogenous (**E**) H3. (**D**) Mass spectrometry (MS) analysis of cumulative H3K27, H3K36, and H3K37 modifications on H3.1 and H3.3. All measured spectra corresponding to H3.1 and H3.3 peptides from both IPs were used for analysis. (**E**) Relative abundance of H3K27, H3K36, and H3K37 modifications on H3.1 variant analyzed separately from MS data of H3.1 and H3.3 purified nucleosomes (left panel). Relative abundance of H3K27, H3K36, and H3K37 modifications on H3.3 variant analyzed separately from MS data of H3.1 and H3.3 purified nucleosomes (right panel). (**F**) Co-occurrence of H2A variants and H3 marks. Mononucleosomes were immunoprecipitated with the indicated antibodies and analyzed for the presence of H2A variants and H3 marks by western blotting. Original pictures of the gels are provided in *Figure 1—source data 1*, *Figure 1—source data 2* and *Figure 1—source data 3*.

The online version of this article includes the following source data and figure supplement(s) for figure 1:

**Source data 1.**

**Source data 2.** The data contains the orginal images of the gels.

**Source data 3.** The data contains the orginal images of the gels.

**Figure supplement 1.** Biochemical analysis of the association between histone variants and histone marks.

**Figure supplement 1—source data 1.** The data contains the orginal images of the gels.

by MS analysis of nucleosomes pulled down with H2A variant-specific antibodies (*Figure 1—figure supplement 1D*). Thus, by contrast with H2A variants, H3 variants do not necessarily form homo-typic nucleosomes, despite the presence of H3 variant-specific deposition mechanisms (*Loppin and Berger, 2020*; *Nie et al., 2014*; *Probst, 2022*), and they do not associate with a specific H2A variant.

Next, we explored the potential link between H3 modifications and H3 variants. We found that all the methylation and acetylation marks surveyed were present on transgenic (T) and endogenous (E) H3 in both H3.1 and H3.3 immunoprecipitations (*Figure 1C*), although there was some degree of pref-erential association of H3K4me, H3K36me, and H3 acetylation with H3.3 and of H3K27me1 with H3.1 (*Figure 1C*). H3.1 and H3.3 modifications were also analyzed by MS by using information from both transgenic and endogenous H3 in respective immunoprecipitations. This approach unambiguously distinguishes the modifications of peptides covering lysines 27 (H3K27) and 36 (H3K36) originating from the endogenous or the transgenic copies of H3.1 and H3.3 (*Figure 1—figure supplement 1E*). We put our focus on modifications at these two positions, because they are diagnostic of constitu-tive heterochromatin (H3K27me1), facultative heterochromatin (H3K27me3), and actively transcribed euchromatin (H3K36me). In contrast to previous reports (*Johnson et al., 2004*; *Zhang et al., 2007*) we identified similar levels of H3K27me2 and H3K27me3 marks on both H3.1 and H3.3 whereas the constitutive heterochromatin mark H3K27me1 was higher on H3.1 (*Figure 1D and E*; *Figure 1—figure supplement 1E-G*). High levels of H3K27me1 on H3.3 were unexpected considering the speci-ficity of H3K27 methyltransferases ATXR5/6 for H3.1 (*Jacob et al., 2014*) and are the likely product of the demethylation of H3K27me3 by the JUMONJI H3K27 demethylases (*Gan et al., 2015*; *Liu et al., 2010*) or intermediates of methylation by the Polycomb Repressive Complex 2. Acetylation and all three methylation states at H3K36 that are associated with active chromatin were likewise found on both H3.1 and H3.3 but with higher enrichment on H3.3 (*Figure 1D and E*; *Figure 1—figure supple-ment 1E-G*). This is consistent with western blot analyses (*Figure 1C*) and with the established role of H3.3 as a replacement variant during transcription (*Delaney and Almouzni, 2023*). The levels of H3K27 and H3K36 modifications on H3.1 and H3.3 displayed the same trends irrespective of whether they were precipitated with either H3.1 or H3.3 nucleosomes, contrasting with the predominant enrichment of H3K37 modifications on H3.3 (*Figure 1E*). The H3 region covering K4, K9, K14, K18, and K23 residues cannot be distinguished between H3.1 and H3.3 by either bottom-up MS or western blotting. Therefore, we analyzed H3K9me1 and H3K9me2, which mark constitutive heterochromatin in plants, only on transgenic H3 and found that both marks were similarly distributed among H3.1 and H3.3 (*Figure 1—figure supplement 1H*) as reported previously (*Johnson et al., 2004*). Also, acetyl-ation of H3K9, H3K14, H3K18, H3K23 and combinations thereof were very similar in H3.1 and H3.3 immunoprecipitations (*Figure 1—figure supplement 1I-K*). Modifications on H3K4 were not analyzed by MS because our experimental setup produced too short peptides to be captured by nanoLC. As the total numbers of measured spectra covering H3 modifications were nearly identical for H3.1 and H3.3 (*Figure 1—figure supplement 1F*) our MS together with western blot data suggest that the vast majority of H3 modifications are not H3 variant specific to the notable exception of H3K36 and H3K37

modifications that are associated preferentially but not uniquely with H3.3. This preference for H3.3 might be related to the presence of substitutions at position 41 specific that distinguishes H3.1 from H3.3 in plants (*Borg et al., 2021*; *Probst, 2022*). These findings suggested that in the heterotypic nucleosomes the same modifications are found on either H3 variants present irrespective of their identity.

Unlike H3 variants, homotypic nucleosomes containing either H2A.W, H2A.Z, H2A, or H2A.X showed marked enrichment of specific histone modifications (*Figure 1F*). Histone modifications in constitutive heterochromatin (H3K9me1, H3K9me2, and H3K27me1) were primarily associated with H2A.W consistent with their synergistic impact on silencing transposons (*Bourguet et al., 2021*). The modification H3K27me3 (facultative heterochromatin) was detected only in H2A.Z nucleosomes, and H3K36me3 (marking euchromatin) was primarily detected in H2A, H2A.X, and H2A.Z nucleosomes. Other modifications, such as H3K4me1 and H3K4me3, displayed complex patterns of association with H2A variants, among themselves, and with other H3 modifications (*Figure 1F*). Surprisingly, H3K4me1 and H3K4me3 co-precipitated high levels of H3K27me1, H3K27me3, and H3K36me3 and low levels of H3K9me1 and H3K9me2 (*Figure 1F*) showing that repressive and active H3 marks co-exist on the same nucleosomes or potentially on the same H3 tail. Similar complex interplay between H3 marks has also been observed in mouse ES cells (*Schwämmle et al., 2016*). Therefore, H2A but not H3 variants form homotypic nucleosomes that preferentially carry specific histone modifications associated with either the transcriptional status of protein-coding genes or transposons.

## Associations of histone variants with the chromatin states

We performed ChIP-seq using *Arabidopsis* seedlings to identify the combinations of all histone variants present in somatic cells and their associations with twelve prominent histone modifications. We included the most abundant isoforms of H2A.W (H2A.W.6 and H2A.W.7), H2A.Z (H2A.Z.9 and H2A.Z.11), and H2A (H2A.2 and H2A.13) in our analysis. We used specific polyclonal antibodies to detect each H2A variant (see *Figure 2—figure supplement 1*) for the demonstration of antibodies against H2A.Z.11 and H2A.2 and (*Osakabe et al., 2021*; *Osakabe et al., 2018*). The algorithm ChromHMM (*Ernst and Kellis, 2017*) was used to define chromatin states in *Arabidopsis* (see Methods for details). We chose to analyze the 26 chromatin states model, which proved to be most parsimonious (*Figure 2—figure supplement 2A*; see Methods). Chromatin states were clustered based on the emission probability for each modification and histone variant (*Figure 2A*) and were distinguished based on distinct combinations of enrichment amongst the 27 chromatin components analyzed (*Figure 2A*). Chromatin states occupied different proportions of the genome (*Figure 2—figure supplement 2B*) and showed distinct relative abundance of transposons, repeats, and elements of protein-coding genes (*Figure 2B*). For each chromatin state, we measured the average level of transcriptional activity (*Figure 2C*), the degree of enrichment in CG methylation (*Figure 2D*)**,** the degree of enrichment in CHG and CHH methylation (*Figure 2—figure supplement 2C, D*), the level of accessibility by DNase I-seq (*Figure 2E*), the nucleosome density determined by MNase-seq (*Figure 2—figure supplement 2E*), and the length in base pairs (*Figure 2—figure supplement 2F*). Chromatin states H1–H6 showed low accessibility, the highest levels of DNA methylation, and primarily occupied transcriptionally inactive transposons and repeats, thus representing constitutive heterochromatin. The states F1–F6 were associated with low transcriptional activity and were present over genes and pseudogenes, indicating that they represented facultative heterochromatin. As defined by state occupancy, constitutive and facultative heterochromatin composed 17% and 20% of the genome, respectively (*Figure 2—figure supplement 2B*), corresponding to previous estimates (*Roudier et al., 2011*). The states E1–E11 occupied expressed genes and thus comprised euchromatin. Three states (I1–I3) showed the lowest nucleosome density and highest accessibility and occupied a large fraction of non-coding RNAs (*Figure 2—figure supplement 2G*) and a quarter of untranslated regions of genes (3'UTR and 5'UTR) (*Figure 2B*). Hence, the states I1–I3 were classified as intergenic. We conclude that specific groups of chromatin states define constitutive heterochromatin (H1-6), facultative heterochromatin (F1-6), euchromatin (E1-11), and intergenic regions (I1-3).

Overall, the chromatin states recapitulate the preferential associations between H2A variants and histone modifications observed in mononucleosomes (*Figure 1*; *Figure 1—figure supplement 1*). H2A.W and H3K9me1 are the determining marks of all six heterochromatic states. H2A.Z, with the polycomb histone modifications H2AK121ub and H3K27me3, are the hallmark of facultative

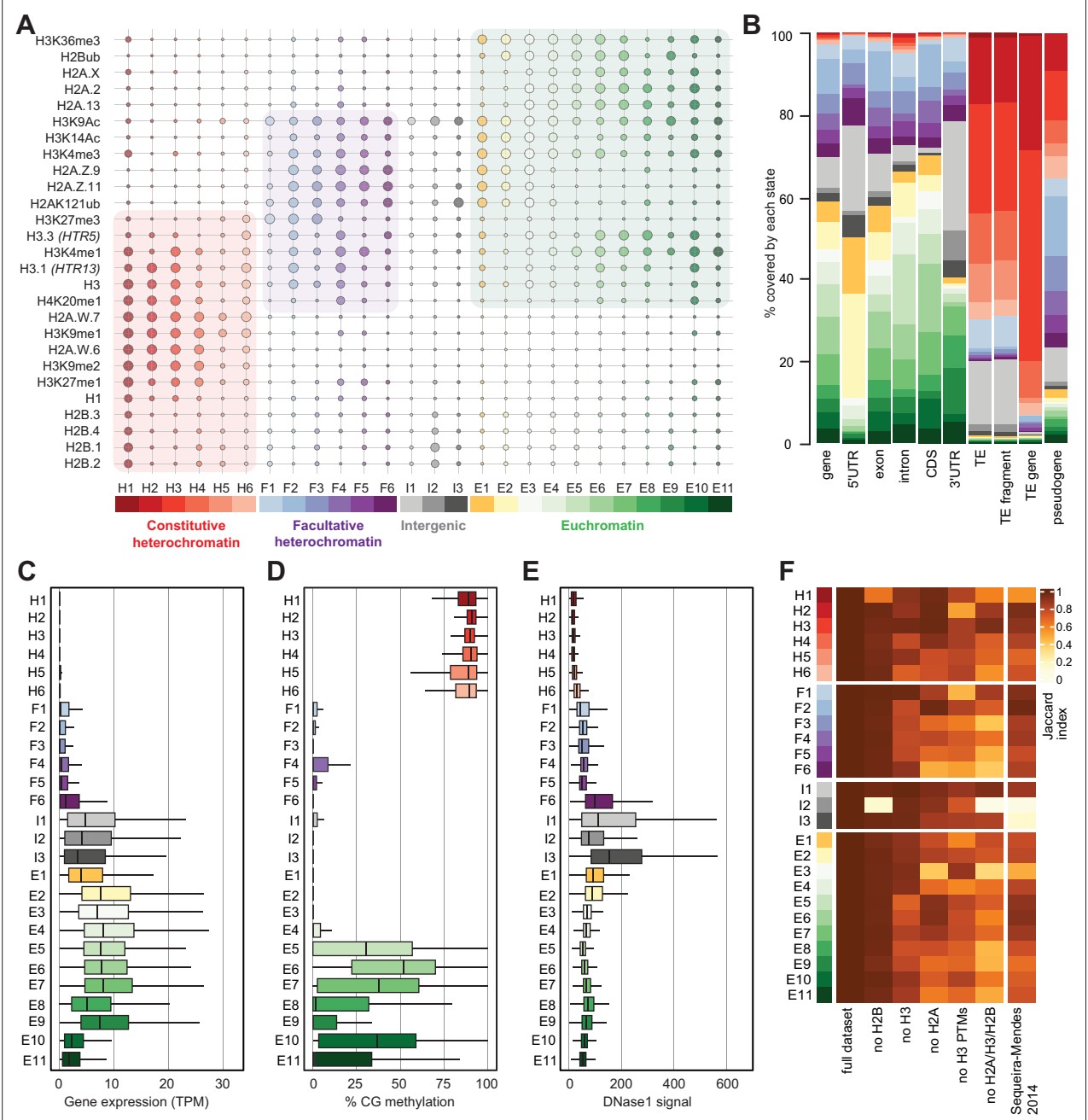

**Figure 2.** Histone variants define chromatin states in *Arabidopsis thaliana*. (**A**) Bubble plot showing the emission probabilities for histone modifications/variants across the 26 chromatin states. (The size of the bubble represents the emission probability ranging from 0 to 1). The colors are ascribed for each type of chromatin. (**B**) Stacked bar plot showing the overlap between annotated genomic features and chromatin states. (**C**) Box plot showing the expression of protein-coding genes overlapping with each chromatin state in Transcripts per Million (TPM). (**D**) Box plot showing levels of CG methylation across chromatin states. (**E**) Box plot comparing DNase I-seq read coverage across chromatin states representing chromatin accessibility. (**F**) Heatmap showing the Jaccard similarity index between the states generated using the whole model and states using a subset of marks, i.e., excluding a set of marks and variants as indicated on the x-axis. The comparison with a 9-state model (*Sequeira-Mendes et al., 2014*) did not include CG content, DNA methylation, H4K5ac, and H3K4me2 which were not used in the 26-state model. H2B did not seem to make a significant contribution. Excluding H3 did reduce effectively the Jaccard index for both the 26 and the 9 chromatin states model, which both included H3.1 and H3.3. The major differences between the 26 and the 9 chromatin states model are in the intergenic states. Overall H2A variants affected most strongly the Jaccard index.

The online version of this article includes the following source data and figure supplement(s) for figure 2:

**Source data 1.** This source data contain the original pictures of the gels.

**Figure supplement 1.** Validation of H2A.2 and H2A.

*Figure 2 continued on next page*

*Figure 2 continued*

**Figure supplement 1—source data 1.** This source data contain the original pictures of the gels.

**Figure supplement 2.** Properties of chromatin states in *Arabidopsis thaliana*.

heterochromatin states. H2A and H2A.X associated with H2Bub and H3K36me3 mark euchromatic states. Contrasting with the strong association between H2A variants and the three major domains of chromatin, much less prominent associations are observed between histone modifications and either H3 or H2B variants.

To examine the importance of H2A variants in the definition of chromatin states, we compared the 26 chromatin states defined in our study with the 9 states defined in a previous study that did not include the comprehensive set of histone variants present in *Arabidopsis* chromatin (*Sequeira-Mendes et al., 2014*; *Figure 2—figure supplement 2H*). The blocks of heterochromatic states H1–H6 corresponded to the previously identified states 8 and 9, which were also defined as heterochromatin in a previous study (*Sequeira-Mendes et al., 2014*). States F1–F6 tended to map to the previously defined facultative heterochromatin states 5 and 6, although state F6 was split among the three states 2, 4, and 6. Similarly, although there was a broad correspondence between euchromatin states E1– E11 with states 1, 3, and 7, there were noticeable differences. Overall, several newly defined states were not associated with the corresponding types of chromatin domains described in previous studies. Previously defined states 2, 4, and 6 contained elements belonging to multiple newly defined chromatin states. To test the dissimilarity between the 9-state and the 26-state models we calculated a 26-state model based on the chromatin marks used to define the 9-state model. The difference between the resulting matrices were measured by the Jaccard index (*Figure 2F*). This showed an overall coherence between the two models, with only mis-assignment of chromatin identity by the 9-state model of chromatin in intergenic regions, H1, H5, H6, and a lower diversity of euchromatin states. Altogether, the addition of the comprehensive set of histone variants in the 26-state model provides a more refined and coherent classification of elements of chromatin than the 9-state model defined primarily based on chromatin modifications and the variants H3.1 and H3.3, suggesting the importance of histone variants in the definition of chromatin states.

To evaluate the contribution of individual components in defining the chromatin states, we recalculated chromatin states after excluding either H2B variants, H3 variants, H2A variants, H3 modifications (no H3 PTMs) or all histone variants (no H2A/H3/H2B) from the dataset used to learn the 26-state model. The difference between the resulting matrices of chromatin states was measured by the Jaccard index (*Figure 2F*). Removing H2B variants did not affect most chromatin states, except for H1 and I2, which showed high emissions probabilities for H2B variants (*Figure 2A*). In contrast, H3 variants showed larger contributions in defining specific states, and removing all H2A variants caused the loss of several states marked by a Jaccard index <0.7 (*Figure 2F*). Eventually, removing all histone variants had a stronger impact than removing all H3 modifications from the model. In summary, our computational and biochemical analyses led to conclude that, among histone variants, H2A variants have the strongest effect on defining the chromatin states by their association with histone modifications.

## Perturbation of H2A variant dynamics affects the genomic localization of chromatin states

We considered the idea that the dynamics of the exchange of H2A variants would affect the chromatin states. To test this idea, we studied the impact of the loss of the chromatin remodeler DDM1, which binds and controls the deposition of H2A.W (*Osakabe et al., 2021*), on the definition of chromatin states. In the *ddm1* mutant, TEs show a loss of DNA methylation, H3K9 methylation, and linker histone H1 as well as an increase in H3K27me3 (*Jeddeloh et al., 1998*; *Kakutani et al., 1999*; *Osakabe et al., 2021*; *Rougée et al., 2021*). This suggested that the loss of H2A.W deposition had a profound impact on chromatin states.

In our previous study, we used the leaves of five-week-old plants to show the impact of *ddm1* on the profiles of H2A.W.6, H2A.X, H1, H3K9me2, H3K36me3 and H3K27me3 (*Osakabe et al., 2021*). This study showed that DDM1 causes the deposition of H2A.W.6 to heterochromatin and we thus extended this investigation to the two other marks of heterochromatin H3K9me1, H2A.W.7 and

H2A.Z.9 and H2A.Z.11 in leaves (11 profiles in total) of *ddm1* and wild-type. Because these profiles were acquired on leaves while seedlings were used to define the 26-state model in the wild-type, we first considered whether the development stage could affect the definition of chromatin states in the wild-type. To test this idea, we built a concatenated chromHMM model based on the profiles of the same set of histone variants and modifications from 10-day-old seedlings and leaves of five-week-old plants (*Figure 3—figure supplement 1A*). Although this model had only 15 chromatin states we observed an association between H2A variants and histone modifications comparable to the model describing the 26-state model (*Figure 2A*). The absence of most typical euchromatic histone modifications in the model likely explained the strong reduction of the number of states of euchromatin compared with the 26-state model (*Figure 3—figure supplement 1A*). The complexity of the different heterochromatin states were preserved in the concatenated model, supporting that the set of H2A variants and histone modifications used to study *ddm1* are sufficient to represent the complexity of the heterochromatin affected directly by *ddm1*. The composition of the chromatin states did not vary significantly between seedlings and leaves and each state occupied a similar proportion of the genome in seedlings or leaves to the exception of state 5 present primarily in leaves and state 13 only present in seedlings (*Figure 3—figure supplement 1A* right column with green bars). However, as expected by the dissimilar transcriptomes of these two developmental stages euchromatic states occupied distinct genes in seedlings and leaves (*Figure 3—figure supplement 1B*). This indicated that neither tissue type nor developmental stage has a dramatic effect on the definition of chromatin states in the wild-type.

To directly compare chromatin states between *ddm1* and wild-type, we built a concatenated chromHMM model (*Ernst and Kellis, 2017*). The concatenated model created a common set of 16 chromatin states that are shared between wild-type and *ddm1* (*Figure 3A*) and identified the percentage of the genome occupied by each chromatin states for each genotype individually (see Methods) (*Figure 3A*, and bar plot on the left). The states were classified as constitutive heterochromatin (hI-hIII), facultative heterochromatin (fI-fVI), euchromatin (eI-eIII), and intergenic (iI) (*Figure 3A*), based on their emission probability and genomic overlap with states from those groups in the 26-state model (see Methods). The presence of chromatin profiles from *ddm1* caused three additional mixed states in the 16-state model, which represents regions covered by facultative and constitutive heterochromatin (fhI) or by a combination of intergenic and constitutive heterochromatin (ihI,ihII) in the 26-state model. Because several euchromatic histone PTMs were not included in the new model, only three states represented euchromatin (eI-eIII) compared with the 11 euchromatic states in the 26-state model. In the 16-state concatenated model with *ddm1* the complexity of facultative (fI-fVI) was preserved but the complexity of heterochromatin states was reduced to three (hI-hIII) compared with the six heterochromatin states in the wild-type 26-state model. Overall, *ddm1* caused a perturbation of the chromatin states associated with constitutive heterochromatin and did not significantly affect the group similarity for facultative heterochromatin and euchromatin states between the 16- and 26- state models, with genomic overlaps between 68 and 100% for the non-mixed states (*Figure 3—figure supplement 1C*).

To compare the chromatin state assignments between *ddm1* and wild-type genomes, we measured the Jaccard index (*Figure 3B*; *Figure 3—figure supplement 1D*) and overlap (*Figure 3C*) of each state between the wild-type and *ddm1*. The strongest differences, in terms of genomic coverage, between *ddm1* and wild-type were observed in constitutive and facultative heterochromatin (*Figure 3D*). Accordingly, the regions marked by heterochromatin (red states h) became covered by states typical of facultative heterochromatin (blue states f) or intergenic states (gray states iI, ihII) (*Figure 3—figure supplement 1C and E*). To a lesser extent, regions covered by facultative state fVI and euchromatic state eIII were converted to intergenic state iI. Hence, the loss of DDM1 broadly affected the association of chromatin states with regions covered with constitutive heterochromatin in the wild-type. Accordingly, chromatin states were changed over TEs, including TE fragments and TE genes (assigned to TE families in TAIR10 annotation) but did not change over protein-coding genes (*Figure 3—figure supplement 1F*). Overall, the constitutive heterochromatin present over TEs in wild-type was replaced in *ddm1* by chromatin states found in intergenic, facultative heterochromatin, and euchromatin in wild-type (*Figure 3E and F*; *Figure 3—figure supplement 1E, F*). In addition, we observed that *ddm1* affected not only constitutive heterochromatin but also regions of euchromatin and facultative heterochromatin that adopted chromatin states distinct from these found in the wild-type. suggesting

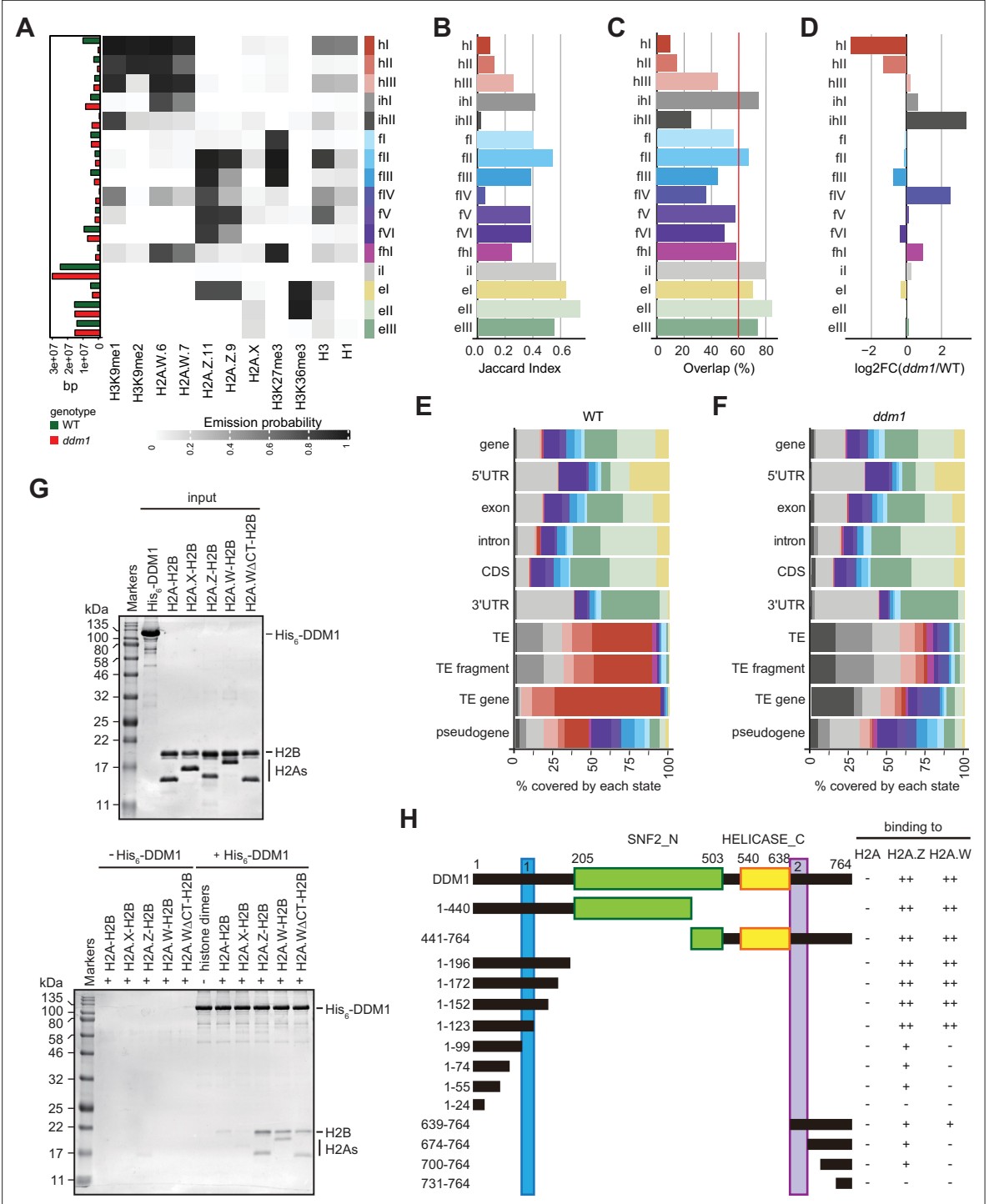

**Figure 3.** Decreased in DNA Methylation (DDM1) loss of function disrupts chromatin states in *Arabidopsis thaliana*. (**A**) Heatmap showing the emission probability for each mark/variant across the 16 chromatin states of the concatenated wild-type and *ddm1* mutant model. The bar plot on the left represents the proportion of the genome covered by each state in wild-type (green) and in *ddm1* (red). (**B**) Bar plot showing the Jaccard indices between the state assignments in wild-type and *ddm1* mutant. (**C**) Bar plot showing the state assignment overlap between the wild-type and *ddm1* for each chromatin state. The red vertical line represents the genome-wide overlap (62.2%). (**D**) Bar plot showing the log2 fold changes of the proportion of genome covered by each state across the *ddm1* genome compared to the wild-type. (**E**) Stacked bar plot showing the overlap between annotated genomic features and chromatin states from the concatenated model in wild-type. (**F**) Stacked bar plot showing the overlap between annotated genomic features and chromatin states from the concatenated model in *ddm1* mutant. (**G**) DDM1 interaction with H2A.W and H2A.Z. Coomassie-stained 15% SDS-PAGE gel showing input protein samples (top panel) used for in vitro pull-down (bottom panel) with His₆-tagged DDM1 and histone

*Figure 3 continued on next page*

**Figure 3 continued**

dimers. The lane ΔCTH2A.W shows that the deletion of the C-terminal tail of H2A.W does not influence binding to DDM1. (**H**) Summary of the pull-down assays to identify regions in DDM1 binding to H2A.W and H2A.Z. Blue and purple boxes indicate the H2A.W binding regions in DDM1 identified by previous work (*Osakabe et al., 2021*). Original pictures of the gels are provided in *Figure 3—source data 1*.

The online version of this article includes the following source data and figure supplement(s) for figure 3:

**Source data 1.** The data contains the orginal images of the gels.

**Figure supplement 1.** Decreased in DNA Methylation (DDM1) loss of function disrupts chromatin states in *Arabidopsis thaliana*.

**Figure supplement 2.** Interaction of Decreased in DNA Methylation (DDM1) and DDM1 deletion mutants with histone variants.

**Figure supplement 2—source data 1.** The souce data file contain the original pcitures of the gels.

indirect effects of the disruption of constitutive heterochromatin caused directly by the loss of DDM1 (*Figure 3—figure supplement 1F*). We thus concluded that the loss of DDM1 causes a profound perturbation of both the definition of chromatin states and their distribution.

## The H2A.W-binding domains of DDM1 bind H2A.Z

The conversion of chromatin states occupied by H2A.W in wild-type to chromatin states occupied by other H2A variants in *ddm1* suggested that DDM1 could also control the dynamics of other H2A variants. We have previously shown that DDM1 specifically binds H2A.W but neither H2A.X nor H1 linker histones (*Osakabe et al., 2021*). To determine if DDM1 binds to H2A or H2A.Z in addition to H2A.W, we performed a Ni-NTA agarose pulldown assay with recombinant histone heterodimers composed of H2B and one of the four H2A variants. We found that His$_6$-tagged DDM1 co-precipitated heterodimers containing H2A.W and H2A.Z but not H2A or H2A.X (*Figure 3G*). This demonstrated that DDM1 bound H2A.W and H2A.Z but not H2A and H2A.X. Notably, the C-terminal tail of H2A.W was dispensable for DDM1 binding (*Figure 3G*). To test whether the conserved motifs involved in H2A.W binding also bind H2A.Z, we prepared a series of DDM1 fragments fused with either His$_6$- or GST-tag. Our pulldown assays showed that fragments containing residues 100–123 or 639–673 were able to bind H2A.W and H2A.Z, whereas some additional regions of DDM1 showed weak binding only to H2A.Z (*Figure 3H*; *Figure 3—figure supplement 2A-C*). Thus, DDM1 uses the same conserved sites to bind H2A.Z and H2A.W.

## Exchange of chromatin states defined by H2A.W and H2A.Z is not directly associated with transcription

H2A.W evolved in land plants independently from macroH2A in animals (*Osakabe and Molaro, 2023*). macroH2A is also associated with heterochromatin although not as strictly as H2A.W (*Sun and Bernstein, 2019*). The deposition of macroH2A depends at least in part on the DDM1 murine ortholog LSH (*Ni et al., 2020*), and it was proposed that LSH binds and exchanges H2A to macroH2A (*Ni and Muegge, 2021*). Finding that DDM1 uses the same sites to bind both H2A.Z and H2A.W suggested that DDM1 controls the deposition of H2A.W and H2A.Z. To test this hypothesis, we surveyed the enrichment of H2A variants and histone PTMs over TEs in *ddm1* and compared this with the wild-type. To prevent carry-over mutations from generations of self-fertilized *ddm1* homozygous mutants we used leaves from *ddm1* homozygous mutant plants segregated from heterozygous *ddm1* mutants. We observed an overall enrichment of H2A.Z over TE genes in *ddm1*, supporting the idea that DDM1 promotes the removal of H2A.Z at TE genes where DDM1 deposits H2A.W in the wild-type (*Figure 4A*). Remarkably, non-transcribed TE genes also accumulated H2A.Z (*Figure 4A*) supporting the conclusion that the replacement of H2A.W by H2A.Z observed in *ddm1* mutants is not governed by transcription. Transcribed TE genes showed enrichment in H2A.Z at their TSS (*Figure 4A*) but also H2A.X on the gene body (*Figure 4—figure supplement 1A*). Transcription of TEs in *ddm1* did not affect the degree of enrichment for H3K27me3 of facultative heterochromatin but with increasing levels of transcription, there was increased enrichment of H3K36me3, resulting in a chromatin profile typical of expressed protein-coding genes (*Figure 4—figure supplement 1B*). Accordingly, strongly expressed TE genes showed an accumulation of euchromatic states, in contrast with non-expressed TE genes that became covered with states typical of facultative heterochromatin, irrespective of the type of constitutive heterochromatin observed on these TE genes in wild-type (*Figure 4B and C*; *Figure 4—figure supplement 2A*). The change of chromatin states did not correlate with the length

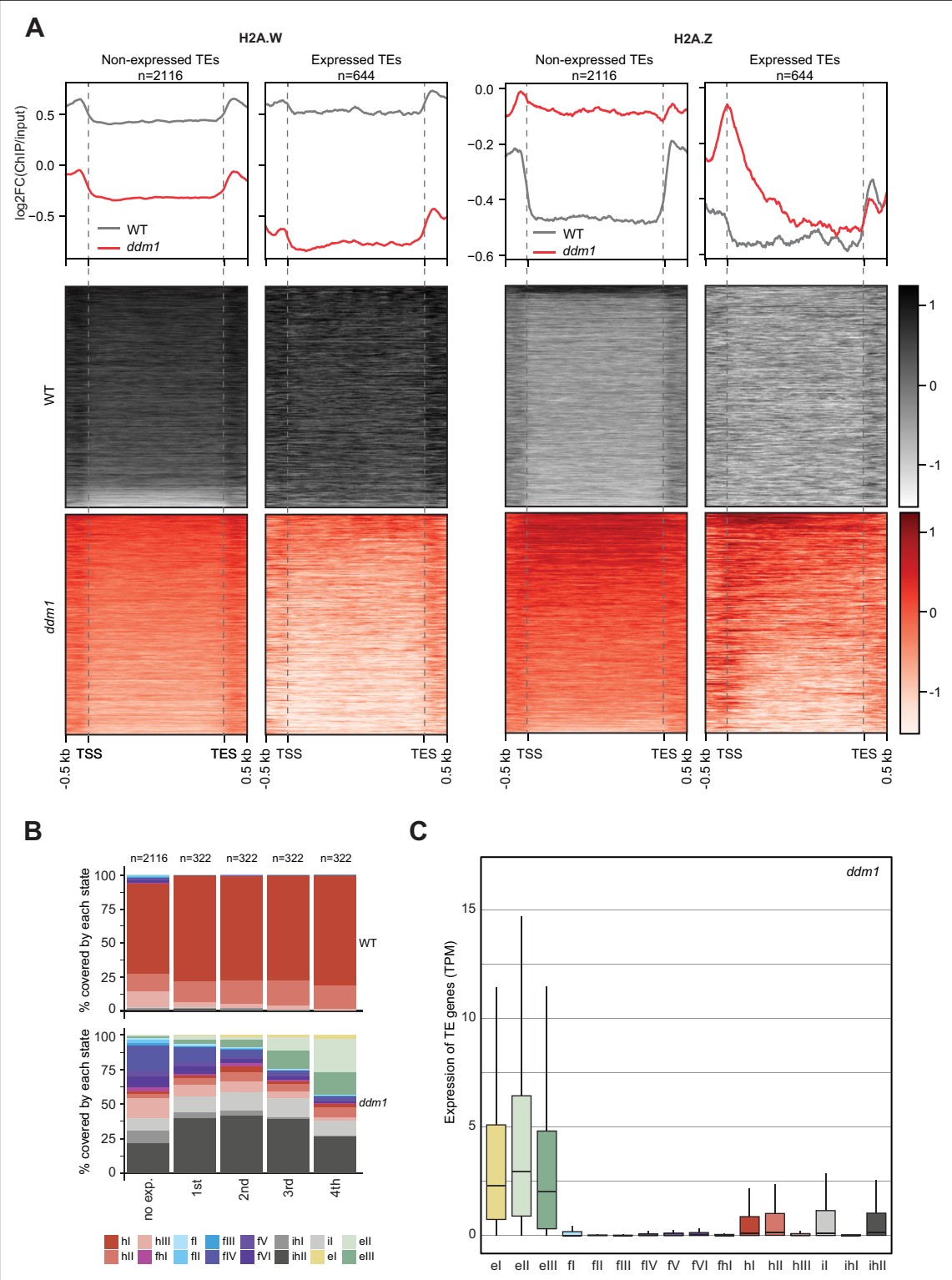

**Figure 4.** Impact of expression on chromatin states over TE genes in *ddm1*. (**A**) Enrichment profiles of H2A.W.6 and H2A.Z.9 over TE genes in *ddm1*. TE genes were grouped by expression in *ddm1* mutant. Out of the 3901 TE genes in the *Arabidopsis* genome annotation, 497 were excluded because they showed expression in wild-type, 2116 TE genes showed no expression in *ddm1* (non-expressed TE genes) while 1288 TE genes were expressed. Because many of these TE genes showed very low expression levels, we divided the expressed TEs into 4 quartiles (322 TE genes each) based on their expression values where the 1st quartile contains TE genes with the lowest expression and the 4th quartile contains TE genes with the highest expression. Given that the TE genes in 1st and 2nd quartile showed nominal expression values, we placed only TE genes in 3rd and 4th quartile (644 TE

*Figure 4 continued on next page*

*Figure 4 continued*

genes) in the category of expressed TEs. n represents the number of TE genes in each group. (**B**) Stacked bar plots of the proportion of states in wild-type (top panel) and in *ddm1* (bottom panel) overlapping TE genes grouped by expression in *ddm1*. (**C**) Box plot showing the expression of TE genes overlapping the 16 concatenated model states in *ddm1*.

The online version of this article includes the following figure supplement(s) for figure 4:

**Figure supplement 1.** Analyses of the parameters that could correlate with the chromatin states with TE expression in *ddm1*.

**Figure supplement 2.** Analyses of the parameters that could correlate with the chromatin states with TE expression in decreased in DNA Methylation (*ddm1*).

of the TE gene (*Figure 4—figure supplement 2B*) or the homogeneity of the chromatin landscape (*Figure 4—figure supplement 2C*). We conclude that at TE genes, DDM1 prevents the replacement of H2A.W by H2A.Z in a transcription-independent manner. Whether the replacement of H2A.W by H2A.Z in *ddm1* is caused by a direct exchange comparable to the exchange of H2A by H2A.Z catalyzed by SWR1 (*Ranjan et al., 2020*) or is the result of a cascade of activities of chaperones and remodelers is unclear. Yet, the alteration in the dynamics of H2A.W and H2A.Z in *ddm1* perturbs the allocation of chromatin states at constitutive heterochromatin and in other regions of chromatin. In some TEs, the new chromatin state acquired in *ddm1* permits transcription, which is then accompanied by a further exchange of H2A.Z to H2A and H2A.X and further loss of marks of constitutive heterochromatin and H1 and gain of marks of euchromatin (*Figure 4—figure supplement 1*).

## Discussion

We report that histone modifications and histone variants combine into chromatin states that subdivide the intergenic space, the non-protein-coding space, and the protein-coding genic space of the genome into biologically significant subunits. These chromatin states summarize the prevalent organization of chromatin across cell types and stages of vegetative development from seedlings to mature leaves. Our data suggest that minor variations in chromatin states occur between vegetative cell types. We predict that highly specialized cell types such as gametes (*Borg et al., 2020*; *Borg et al., 2021*) with specific histone variants and unusually high or low levels of chromatin regulators harbor specific chromatin states that are not reflected in our study. Calculation of chromatin states and biochemical analyses demonstrate the strongest associations between nucleosomes homotypic for H2A variants and histone modifications while nucleosomes are primarily heterotypic for H3 variants with a weaker association with histone modifications.

The prominent role of H2A variants in the organization of chromatin states is supported by systematic evaluation of the individual contribution of histone variants, H3 marks, and a combination thereof (*Figure 2F*) and the impact of the loss of the remodeler DDM1, which deposits H2A.W to maintain TE silencing (*Osakabe et al., 2021*). We further show that the DDM1 binding sites for H2A.W also bind H2A.Z, but not H2A or H2A.X. In *ddm1*, we observe a net replacement of H2A.W by H2A.Z. The effect can be interpreted in several manners. DDM1 could exchange directly H2A.Z to H2A.W. This model would support the idea of the convergent evolution of DDM1 and its orthologs LSH and HELLS in mammals with LSH catalyzing the exchange between replication dependent H2A and macroH2A (*Berger et al., 2023*; *Ni and Muegge, 2021*). An alternative model is that the replacement of H2A.W by H2A.Z is indirect in *ddm1*. For example, H2A.Z might be deposited because of the loss of DNA methylation in *ddm1*, which is consistent with the well-established mutually exclusive patterns of H2A.Z and DNA methylation (*Zilberman et al., 2008*). It is also possible that the binding of DDM1 to H2A.Z is relevant in a different context that is not illustrated by our study. Whichever mechanism is involved it is unlikely that on its own, the loss of H2A.W in *ddm1* causes the deposition of H2A.Z on constitutive heterochromatin because this is not observed in mutants lacking H2A.W (*Bourguet et al., 2021*). The resulting enrichment of H2A.Z in *ddm1* is accompanied by a corresponding exchange of constitutive to facultative chromatin states over TEs. Notably, this does not require transcription, as TEs that remain silent in *ddm1* are likewise occupied by a facultative heterochromatin marked by H2A.Z and H3K27me3. When TEs are strongly expressed in *ddm1*, they also acquire euchromatin states marked by H2A and H2A.X over their gene body, while H2A.Z remains confined to the transcription start site. Hence, H2A variants appear to be important in shaping the deposition of histone modifications and defining chromatin states.

Overall, our observations suggest that chromatin writers and erasers operate with different affinities and efficacies based on the H2A variants in the nucleosome, an idea that is supported by previous reports. The inhibition of the H2B ubiquitin ligase by the H2A.Z tail (*Surface et al., 2016*) is sufficient to explain the anticorrelation between H2Bub and H2A.Z over gene bodies. H2A.Z is the major substrate for H2AK121Ub deposited by PRC1, as indicated by their co-occurrence within chromatin states as well as previous biochemical analyses (*Gómez-Zambrano et al., 2019*). PRC2 strongly prefers arrays of H2A.Z as a substrate (*Wang et al., 2018*), thus supporting the link between H2A.Z and H3K27me3 (*Carter et al., 2018*). We propose to expand these examples to the more general idea that the nature of the H2A variant controls the activity of the machinery that either deposits or erases the histone modifications. Such a model would explain the importance of histone variants in the establishment of chromatin states.

How are chromatin states maintained? In mammalian cells, there is a broad maintenance of the patterns of H3 modifications by recycling H3 and H4 at the replication fork (*Reverón-Gómez et al., 2018*). As plant cells divide, H3.1 sustains the deposition of H3K27me3 (*Jiang and Berger, 2017*) and H3K27me1 (*Jacob et al., 2014*) thus providing a positive feedback loop to sustain the polycomb repressive state and constitutive heterochromatin in dividing cells, respectively. Recent data in mammalian cells show that the recycling of H2A-H2B after replication provides another positive feedback loop to maintain the patterns of H2A variants (*Flury et al., 2023*). Such replication-dependent maintenance mechanisms no longer operate in non-dividing cells, providing opportunities for changing the allocation of chromatin states. This is illustrated by several studies in *Arabidopsis*. The positive feedback loops that maintain histone modifications become inactive and reversion to active chromatin states becomes possible by the deposition of H3.3 and eviction of H2A.Z by INO80 in response to temperature (*Zhao et al., 2023*) or to light (*Willige et al., 2023*). Mechanisms coupling H3.3 and H2A.Z dynamics are likely general for the regulation of responsive genes that are covered with facultative chromatin states (*Long et al., 2023*). Conversely, on expressed genes, acetylated histones assist in the docking of the chromatin remodeler SWI2/SNF2-related 1 (SWR1), which deposits H2A.Z at the transcription start sites (TSS) of expressed genes (*Aslam et al., 2019*; *Bieluszewski et al., 2022*).

Although H3.1 and H3.3 do not strongly differentiate the three main classes of chromatin, they likely refine their definition. A relative enrichment of H3.1 with constitutive heterochromatin modifications (*Johnson et al., 2004*) were previously noted and confirmed in our study. Accordingly, the preferred substrate for the deposition of the heterochromatic mark H3K27me1 is H3.1 (*Jacob et al., 2014*). We also observed a relationship between H3.3 and H3K36me1/2, pointing to a possible preference of H3.3 for the deposition of H3K36me or for the demethylation of H3K36me3. In addition, it is becoming apparent that the dynamics of H3.3 and H2A.Z are coordinated in *Arabidopsis* (*Zhao et al., 2023*) and mammals (*Yang et al., 2022*). We thus propose that as yet unknown mechanisms link the deposition of H2A and H3 variants, resulting in specific types of nucleosomes that are sufficient to orchestrate the deposition of PTMs in distinct chromatin states. These will be read, interpreted, and further modified by silencing mechanisms in heterochromatin and the transcriptional machinery in euchromatin, leading to a changing chromatin landscape responding to the activity of the cell and its responses to the environment.

## Methods

**Key resources table**

| Reagent type (species) or resource | Designation | Source or reference | Identifiers | Additional information |
|---|---|---|---|---|
| Antibody | anti-H2A.X.3/5 (Rabbit polyclonal) | *Yelagandula et al., 2014* | | WB: 1 µg/ml 5 µg per ChIP 10 µg per IP |
| Antibody | anti-H2A.13 (Rabbit polyclonal) | *Yelagandula et al., 2014* | | WB: 1 µg/ml 5 µg per ChIP |
| Antibody | anti-H2A.2 (Rabbit polyclonal) | This work (See Materials and methods) | | WB: 1 µg/ml 5 µg per ChIP |

*Continued on next page*

*Continued*

| Reagent type (species) or resource | Designation | Source or reference | Identifiers | Additional information |
|---|---|---|---|---|
| Antibody | anti-H2A.Z.9 (Rabbit polyclonal) | *Yelagandula et al., 2014* | | WB: 1 µg/ml 5 µg per ChIP 10 µg per IP |
| Antibody | anti-H2A.Z.11 (Rabbit polyclonal) | This work (See Materials and methods) | | WB: 1 µg/ml 5 µg per ChIP |
| Antibody | anti-H2A.W.6 (Rabbit polyclonal) | *Yelagandula et al., 2014* | | WB: 1 µg/ml 5 µg per ChIP 10 µg per IP |
| Antibody | anti-H2A.W.7 (Rabbit polyclonal) | *Lorković et al., 2017* | | WB: 1 µg/ml 5 µg per ChIP 10 µg per IP |
| Antibody | anti-H3 (Rabbit polyclonal) | Abcam | Cat# ab1791, RRID:AB_302613 | WB: 0,5 µg/ml 5 µg per ChIP |
| Antibody | anti- H3K36me3 (Rabbit polyclonal) | Abcam | Cat# ab9050, RRID:AB_306966 | WB: 1 µg/ml 5 µg per ChIP 10 µg per IP |
| Antibody | anti-H3K27me3 (Rabbit polyclonal) | Millipore | Cat# 07–449, RRID:AB_310624 | WB: 1 µg/ml 5 µg per ChIP |
| Antibody | anti-H3K4me3 (Rabbit polyclonal) | Abcam | Cat# ab8580, RRID:AB_306649 | WB: 1 µg/ml 5 µg per ChIP 10 µg per IP |
| Antibody | anti-H3K4me1 (Rabbit polyclonal) | Abcam | Cat# ab8895, RRID:AB_306847 | WB: 1 µg/ml 5 µg per ChIP 10 µg per IP |
| Antibody | anti-H3K27me1 (Rabbit polyclonal) | Millipore | Cat# 17–643, RRID:AB_1587128 | WB: 1 µg/ml 5 µl per ChIP 20 µl per IP |
| Antibody | anti-H1 (Rabbit polyclonal) | Agrisera | AS11 1801 | 5 µg per ChIP |
| Antibody | anti-H4K20me1 (Rabbit polyclonal) | Abcam | Cat# ab9051, RRID:AB_306967 | 5 µg per ChIP |
| Antibody | anti-H3K9me1 (Rabbit polyclonal) | Abcam | Cat# ab8896, RRID:AB_732929 | WB: 1 µg/ml 5 µg per ChIP 10 µg per IP |
| Antibody | anti-H3K9me2 (Mouse monoclonal) | Abcam | Cat# ab1220, RRID:AB_449854 | WB: 1 µg/ml 5 µg per ChIP |
| Antibody | anti-H3K9ac (rabbit polyclonal) | Millipore | Cat# 17–615, RRID:AB_1163437 | WB: 1 µg/ml |
| Antibody | anti-H3K18ac (rabbit polyclonal) | Abcam | Cat# ab1191, RRID:AB_298692 | WB: 1 µg/ml |
| Antibody | anti-H3K23ac (rabbit polyclonal) | Abcam | Cat# ab47813, RRID:AB_880444 | WB: 1 µg/ml |
| Antibody | anti-H3K27ac (rabbit polyclonal) | Abcam | Cat# ab4729, RRID:AB_2118291 | WB: 1 µg/ml |
| Antibody | anti-H3K122ac (rabbit polyclonal) | Abcam | Cat# ab33309, RRID:AB_942262 | WB: 1 µg/ml |
| Antibody | anti-H3K27me2 (rabbit polyclonal) | Abcam | Cat# ab24684, RRID:AB_448222 | WB: 1 µg/ml |
| Antibody | anti-H3K36me2 (rabbit polyclonal) | Abcam | Cat# ab9049, RRID:AB_1280939 | WB: 1 µg/ml |

*Continued on next page*

Continued

| Reagent type (species) or resource | Designation | Source or reference | Identifiers | Additional information |
|---|---|---|---|---|
| Antibody | anti-HA (Rat monoclonal) | Roche | Cat# 11867423001, RRID:AB_390918 | WB: 0,5 µg/ml |
| Antibody | Rabbit IgG (monoclonal) | Abcam | Cat# ab171870, RRID:AB_2687657 | 10 µg per IP |
| Commercial assay or kit | Nugen Ovation Ultralow V2 DNA-Seq library prep kit | NuGen | Cat# 0344 | |
| Commercial assay or kit | NEBNext Ultra II DNA library prep kit for Illumina | New England Biolabs | Cat# E7645L | |
| Commercial assay or kit | Dynabeads Protein A | Invitrogen | Cat# 10746713 | Used for ChIP |
| Commercial assay or kit | Protein A Mag Separose | GE Healthcare (Cytiva) | Cat# 28951378 | Used for IP |
| Commercial assay or kit | Anti-HA Affinity matrix | Roche | Cat# 11815016001 | Used for IP |
| Software, algorithm | ChromHMM | *Ernst and Kellis, 2012*; *Ernst and Kellis, 2017* | RRID:SCR_018141 | |
| Software, algorithm | Trim Galore | DOI:10.5281/zenodo.5127898. | RRID:SCR_011847 | |
| Software, algorithm | Bowtie 2 | *Langmead and Salzberg, 2012* | RRID:SCR_016368 | |
| Software, algorithm | Picard | broadinstitute.github.io/picard/ | RRID:SCR_006525 | |
| Software, algorithm | Deeptools | *Ramírez et al., 2016* | RRID:SCR_016366 | |

## Generation of antibodies, isolation of nuclei, MNase digestion, immunoprecipitation, SDS-PAGE, and western blotting

Antibodies against H2A.Z.11 (KGLVAAKTMAANKDKC) and H2A.2 (CPKKAGSSKPTEED) peptides were raised in rabbits (Eurogentec) and purified by a peptide affinity column. Purified IgG fractions were tested for specificity on nuclear extracts from WT and knock-out lines or with overexpressed histone variants (*Figure 2—figure supplement 1*). Specificities of custom-made polyclonal antibodies against *Arabidopsis* H2A.Z.9, H2A.X, H2A.W.6, H2A.13, H2A.W.7, H2Bs, and linker histone H1 was validated in previous publications (*Jiang et al., 2020*; *Lorković et al., 2017*; *Osakabe et al., 2018*; *Yelagandula et al., 2014*).

For MNase digestion followed by immunoprecipitation, nuclei were isolated from 4 g of 2–3 weeks old leaves and the procedure as described in *Lorković et al., 2017* was followed. Isolated nuclei were washed once in 1 ml of N buffer (15 mM Tris-HCl pH 7.5, 60 mM KCl, 15 mM NaCl, 5 mM MgCl$_2$, 1 mM CaCl$_2$, 250 mM sucrose, 1 mM DTT, 10 mM ß-glycerophosphate) supplemented with protease inhibitors (Roche). After spinning for 5 min at 1800 × *g* at 4 °C nuclei were re-suspended in N buffer to a volume of 1 ml. Twenty microliters of MNase (0.1 u/µl) (Sigma-Aldrich) were added to each tube and incubated for 15 min at 37 °C and during the incubation nuclei were mixed four times by inverting the tubes. MNase digestion was stopped on ice by addition of 110 µl of MNase stop solution (100 mM EDTA, 100 mM EGTA). Nuclei were lysed by the addition of 110 µl of 5 M NaCl (final concentration of 500 mM NaCl). The suspension was mixed by inverting the tubes and they were then kept on ice for 15 min. Extracts were cleared by centrifugation for 10 min at 20,000 × *g* at 4 °C. Supernatants were collected and centrifuged again as above. For each immunoprecipitation extract, an equivalent of 4 g of leaf material was used, usually in a volume of 1 ml. To control MNase digestion efficiency, 100 µl of the extract was kept for DNA extraction. Antibodies, including non-specific IgG from rabbits, were bound to protein A magnetic beads (GE Healthcare) and then incubated with MNase extracts overnight at 4 °C. Beads were washed two times with N buffer without sucrose, containing 300 mM NaCl, followed by three washes with N buffer containing 500 mM NaCl. Beads were re-suspended in 150 µl of 1 × Laemmli loading buffer in 0.2 × PBS.

Proteins were resolved on 15% SDS-PAGE, transferred to a Protran nitrocellulose membrane (GE Healthcare) and analyzed by western blotting using standard procedures. The blots were developed with an enhanced chemiluminescence kit (Thermo Fisher Scientific) and signals acquired by a

ChemiDoc instrument (BioRad). All primary histone variant-specific and H3 marks-specific antibodies were used at 1 μg/ml dilution. H3-specific antibody was used at 1:5000 dilution. Rat anti-HA antibody (Roche 3F10) was used at 1:2000 dilution. Secondary antibodies were goat anti-rabbit IgG (BioRad) and goat-anti rat IgG (Sigma-Aldrich), both at a 1:10,000 dilution.

## Mass spectrometry

For mass spectrometry, immunoprecipitated nucleosomes were resolved on a 4–20% gradient gels (Serva) and silver-stained. Histone H3 bands were excised, reduced, alkylated, in-gel trypsin or LysC digested, and processed for MS. The nano HPLC system used was a Dionex UltiMate 3000 HPLC RSLC nano system (Thermo Fisher Scientific, Amsterdam, Netherlands) coupled to a Q Exactive mass spectrometer (Thermo Fisher Scientific, Bremen, Germany), equipped with a Proxeon nanospray source (Thermo Fisher Scientific, Odense, Denmark). Peptides were loaded onto a trap column (Thermo Fisher Scientific, Amsterdam, Netherlands, PepMap C18, 5 mm 3300 mm ID, 5 mm particles, 100 Å pore size) at a flow rate of 25 ml/min using 0.1% TFA as the mobile phase. After 10 min, the trap column was switched in line with the analytical column (Thermo Fisher Scientific, Amsterdam, Netherlands, PepMap C18, 500 mm 375 mm ID, 2 mm, 100 Å). Peptides were eluted using a flow rate of 230 nl/min and a binary 2 hr gradient. The gradient starts with the mobile phases: 98% solution A (water/formic acid, 99.9/0.1, v/v) and 2% solution B (water/acetonitrile/formic acid, 19.92/80/0.08, v/v/v), increases to 35% of solution B over the next 120 min, followed by a gradient in 5 min to 90% of solution B, stays there for 5 min and decreases in 5 min back to the gradient 98% of solution A and 2% of solution B for equilibration at 30 °C. The Q Exactive HF mass spectrometer was operated in data-dependent mode, using a full scan (m/z range 380–1500, nominal resolution of 60,000, target value 1E6) followed by MS/MS scans of the 10 most abundant ions. MS/MS spectra were acquired using a normalized collision energy of 27%, an isolation width of 1.4 m/z, and a resolution of 30.000, and the target value was set to 1E5. Precursor ions selected for fragmentation (exclude charge states 1, 7, 8, >8) were put on a dynamic exclusion list for 40 s. Additionally, the minimum AGC target was set to 5E3, and the intensity threshold was calculated to be 4.8E4. The peptide match feature was set to preferred, and the exclude isotopes feature was enabled. For peptide identification, the RAW files were loaded into Proteome Discoverer (version 2.1.0.81, Thermo Scientific). The resulting MS/MS spectra were searched against *Arabidopsis thaliana* histone H3 sequences (seven sequences; 951 residues) using MS Amanda v2.1.5.8733 (*Dorfer et al., 2014*). The following search parameters were used: Beta-methylthiolation on cysteine was set as a fixed modification, oxidation on methionine, deamidation of asparagine and glutamine, acetylation on lysine, phosphorylation on serine, threonine, and tyrosine, methylation and di-methylation on lysine and threonine, tri-methylation on lysine, and ubiquitinylation on lysine were set as variable modifications. The peptide mass tolerance was set to ±5 ppm, and the fragment mass tolerance was set to ±15 ppm. The maximal number of missed cleavages was set to 2. The result was filtered to 1% FDR at the peptide level using the Percolator algorithm integrated with Thermo Proteome Discoverer. The localization of the phosphorylation sites within the peptides was performed with the tool ptmRS, which is based on the tool phosphoRS (*Taus et al., 2011*).

Peptides diagnostic for H3.1 and H3.3 covering positions K27 and K36 (see *Figure 1—figure supplement 1E*) was used for the analysis of modifications. All peptides covering these two positions were selected in H3.1 and H3.3 immunoprecipitation samples and analyzed for the presence of modifications with the threshold for modification probability set to 95% or higher. Relative modification levels were expressed as the number of modified peptides (*Figure 1—figure supplement 1G*) divided by the total number of peptides (*Figure 1—figure supplement 1F*) that were measured for each lysine position resulting in total modification levels for H3.1 and H3.3 (see *Figure 1D*). We also analyzed the same data by splitting H3.1 and H3.3 specific peptides in each immunoprecipitation and obtained highly similar trends for H3.1 and H3.3 irrespective of whether they were precipitated with H3.1 or H3.3 (*Figure 1E*). Histone acetylation was analyzed by selecting all peptides covering indicated positions and expressed as relative acetylation levels in each immunoprecipitation without differentiating H3.1 and H3.3 variants (*Figure 1—figure supplement 1I and J*). We also analyzed H3K9, H3K14, and H3K18 acetylation from peptides derived from transgenic copy alone because these data reflect modification levels of each H3 variant and obtained highly similar levels (*Figure 1—figure supplement 1K*) as without differentiation between H3.1 and H3.3.

## Purification of recombinant *Arabidopsis* DDM1 and its truncations

His$_6$-tagged DDM1 and its truncations (DDM1(1–440), DDM1(1–196), DDM1(441–764)), GST-tagged DDM1 truncations, DDM1(1–24), DDM1(1–55), DDM1(1–74), DDM1(1–99), DDM1(1–123), DDM1(1–152), DDM1(1–172), DDM1(1–196), were expressed and purified as described previously (*Osakabe et al., 2021*).

## Purification and reconstitution of recombinant *Arabidopsis* histone heterodimers

To purify and reconstitute *Arabidopsis* histone dimers, recombinant *Arabidopsis* histones H2A.13, H2A.X.3, H2A.Z.9, H2A.W.6, H2A.W.6 ΔCT (aa 1–128), and H2B.9 were expressed and purified as previously described (*Osakabe et al., 2013*; *Tachiwana et al., 2010*; *Tanaka et al., 2004*). The H2A-H2B heterodimers were reconstituted and purified as previously described (*Osakabe et al., 2013*).

## Pull-down assay

The pull-down assay using Ni-NTA and GS4B beads for His$_6$-tagged DDM1 or truncations and GST-tagged DDM1 truncations were performed as described previously (*Osakabe et al., 2021*). Proteins precipitated with beads were analyzed by 15% SDS-PAGE stained with Coomassie Brilliant Blue.

## Plant material for ChIP-seq

Col-0 wild-type (WT) *Arabidopsis thaliana* seeds were stratified at 4 °C in the dark for three days. Seeds were germinated on ½ MS sterilized plates in the growth chamber under long day (LD) conditions (21 °C; 16 hr light/8 hr dark). After 10 days seedling tissue was freshly harvested. For ChIP-seq from leaf tissue, Col-0 wild-type (WT), and seeds from *ddm1*/+ plants were stratified at 4 °C in the dark for three days. To prevent carry-over mutations from generations of *ddm1* homozygous mutants we used leaves from *ddm1* homozygous mutant plants segregated from heterozygous *ddm1*/+ mutants. Plants were grown on soil in the growth chamber under a LD conditions (21 °C; 16 hr light/8 hr dark) and leaf tissue was harvested 5 weeks after germination.

## Chromatin immunoprecipitation (ChIP)

For ChIP nuclei isolation was performed using 10-day-old seedlings (WT Col-0) or leaves (WT Col-0, *ddm1*). The procedure for nuclei isolation and chromatin immunoprecipitation was performed as described in *Osakabe et al., 2021*. Freshly harvested tissues (0.3 g of tissue was used for each immunoprecipitation) were fixed with 1% formaldehyde for 15 min and the cross-linking reaction was stopped by the addition of 125 mM glycine. Crosslinked tissues were frozen and ground with a mortar and pestle in liquid nitrogen to obtain a fine powder. Ground tissues were resuspended in M1 buffer (10 mM sodium phosphate pH 7.0, 100 mM NaCl, 1 M hexylene glycol, 10 mM 2-mercaptoethanol, and protease inhibitor (Roche)), and the suspension was filtered through miracloth. Nuclei were precipitated by centrifugation and washed six times with M2 buffer (10 mM sodium phosphate pH 7.0, 100 mM NaCl, 1 M hexylene glycol, 10 mM MgCl$_2$, 0.5% Triton X-100, 10 mM 2-mercaptoethanol, and protease inhibitor), and then further washed once with M3 buffer (10 mM sodium phosphate pH 7.0, 100 mM NaCl, 10 mM 2-mercaptoethanol, and protease inhibitor). Nuclei pellets were rinsed and resuspended in a sonication buffer (10 mM Tris-HCl pH 8.0, 1 mM EDTA, 0.1% SDS, and protease inhibitor) and sonicated with a Covaris E220 High Performance Focused Ultrasonicator for 15 min at 4 °C (Duty factor 5.0; PIP 140.0; Cycles per Burst 200) in 1 ml Covaris milliTUBE. After chromatin shearing, the debris was removed by centrifugation, and the solutions containing chromatin fragments were diluted with three times the volume of ChIP dilution buffer (16.7 mM Tris-HCl pH 8.0, 167 mM NaCl, 1.2 mM EDTA, 1.1% Triton X-100, 0.01% SDS, and protease inhibitor). After dilution, protein A/G magnetic beads (50 µl for one gram of tissue; Thermo Fisher Scientific) were added to sheared chromatin and incubated at 4 °C for 1 hr with rotation. Pre-cleared samples were collected and incubated with 5 µg of in-house prepared anti-H2A.X.3/5, anti-H2A.13, anti-H2A.2, anti-H2A.Z.9, anti-H2A.Z.11, anti-H2A.W.6, anti-H2A.W.7, anti-H1, and anti-H3 (Abcam, ab1791), anti-H3K36me3 (Abcam, ab9050), anti-H3K27me3 (Millipore, 07–449), anti-H3K4me3 (Abcam, ab8580), anti-H3K4me1 (Abcam, ab8895), anti-H3K27me1 (Millipore 17–643), anti-H4K20me1 (Abcam, ab9051), anti-H3K9me1 (Abcam, ab8896), or anti-H3K9me2 (Abcam, ab1220) antibodies at 4°C overnight with rotation. After incubation, samples were mixed with 30 µl of protein A/G magnetic beads, incubated at 4°C for 3 hr

with rotation, washed twice with low salt buffer (20 mM Tris-HCl pH 8.0, 150 mM NaCl, 2 mM EDTA, 1% Triton X-100, and 0.1% SDS), once with high salt buffer (20 mM Tris-HCl pH 8.0, 500 mM NaCl, 2 mM EDTA, 1% Triton X-100, and 0.1% SDS), once with LiCl buffer (10 mM Tris-HCl pH 8.0, 1 mM EDTA, 0.25 M LiCl, 1% IGEPAL CA-630, and 0.1% deoxycholic acid), and twice with TE buffer (10 mM Tris-HCl pH 8.0 and 1 mM EDTA). Immunoprecipitated DNA was eluted by adding 500 µl of elution buffer (1% SDS and 0.1 M NaHCO$_3$), incubated at 65°C for 15 min, and mixed with 51 µl of reverse crosslink buffer (40 mM Tris-HCl pH 8.0, 0.2 M NaCl, 10 mM EDTA, 0.04 mg/ml proteinase K (Thermo Fisher Scientific)). The reaction mixture was then incubated at 45°C for 3 hr and subsequently at 65°C for 16 hr. After crosslink reversal, DNA was treated with 10 µg of RNaseA (Thermo Fisher Scientific), incubated at room temperature for 30 min, and purified with a MinElute PCR purification kit (Qiagen).

## ChIP-seq library prep and data processing

For ChIP-seq, libraries were prepared either with Nugen Ovation Ultralow V2 DNA-Seq library prep kit (NuGen) or NEBNext Ultra II DNA library prep kit for Illumina (New England Biolabs) following the manufacturer's instructions. The libraries were sequenced with an Illumina Hiseq 2000 to generate single-end 50 bp reads. For alignment and quality check of sequenced samples, bedtools v2.27.1 (*Quinlan, 2014*) was used to convert the raw BAM files to fastq. FastQC v0.11.8 (https://qubeshub.org/resources/fastqc) was used to generate quality reports for all sequencing data. Reads were trimmed using trim_galore v0.6.5 (DOI:10.5281/zenodo.5127898) (*trim_galore --dont_gzip --stringency 1 --fastqc --length 5 $*) and then aligned to the TAIR10 *Arabidopsis* genome using Bowtie2 v2.4.1 (*Langmead and Salzberg, 2012*) with default settings, Picard v2.22.8 (https://broadinstitute.github.io/picard/) was used to remove duplicated reads. Deeptools v3.1.2 (*Ramírez et al., 2016*) was used to examine correlations between the ChIP samples. The bamCompare function from Deeptools was used to normalize ChIP samples to Input or H3 and to generate log2 ratio (ChIP/(Input or H3)) bigwig files.

## ChIP-seq data processing for published data

Publicly available ChIP-seq data for H2B variants were downloaded from GEO GSE151166 (*Jiang et al., 2020*). ChIP-seq data for H3 variants were downloaded from GEO GSE34840 (*Stroud et al., 2012*). ChIP-seq data for H3K14Ac, H3K9Ac was downloaded from GEO GSE89768 (*Kim et al., 2016*). ChIP-seq data for H2AK121Ub and H3K27me3 was downloaded from GEO GSE89357 (*Zhou et al., 2017*). Raw data was downloaded and processed as described above. Quality control metrics for all samples are included in source file 1.

## Defining chromatin states

Aligned ChIP-seq BAM files for chromatin features were generated as described above. Aligned BAM files were then converted to BED format using bedtools bamtobed v2.27.1 (*Quinlan, 2014*). These genome-wide BED files were then used for learning chromatin states by the multivariate Hidden Markov Models. For the extensive wild-type model, we used the BinarizeBed and LearnModel programs from ChromHMM (*Ernst and Kellis, 2017*) with default settings and window size of 200 bp. We generated models from 2 to 50 chromatin states. To compare emission parameters of models with different numbers of states, the command CompareModel from ChromHMM was used. Thus, we selected a reference model with the largest number of states (50) and compared it with other models with lower number of states (from 49 to 2). This resulted in a correlation matrix between each state of the reference model and any state of the other models being compared (*Figure 2—figure supplement 2*). This comparison revealed that the correlation dropped for models including less than 13 states, suggesting that some of the biologically relevant states were not resolved. We thus concluded that at least 13 states should be used. However, ChromHMM (*Ernst and Kellis, 2017*) does not establish the optimal state number as it does not consider genomic features which are associated with specific biological function. Therefore, we analyzed the association between chromatin states with genomic features in all models ranging from 13 to 30 states. As a result, for example, we observed that models with more than 13 states could resolve biologically meaningful heterochromatic states (H2 to H6) (H2 to H4 associated with pericentromeric repeats/CMT2 associated repeats; H5 and H6 associated with chromosome arm repeats/RdDM repeats) (as shown in *Jamge et al., 2022*, Figure 3). Furthermore, only with a 26-state model, we observed a clearly defined novel state H1 with H2B enrichment, largely

deprived of TE genes and located in the closest proximity to centromeres. Increasing the number of states to 27 and above gave rise to an additional chromatin state that could no longer be associated with any distinct genomic feature. Hence, we decided that, with the set of chromatin components included, a 26-state model is optimal for our analysis.

For the mixed seedling and leaf data, and for the mixed wild-type and *ddm1* mutant data we used concatenated ChromHMM models (*Ernst and Kellis, 2017*). Those models use the data from both tissues/genotypes to build a common model. To this end, we used the BinarizeBed and LearnModel with the default settings and window size of 200 bp but with the tissue or genotype as additional information. The number of states in the final model was decided on as described for the extensive model.

## Analysis of sub-epigenome models

The ChromHMM command EvalSubset was used to compare models where histone variants or modifications were excluded from the model. Five models were generated and evaluated against the full model: no H2B variants, no H3 variants, no H2A variants, no histone modifications, and no variants (H2A/H3/H2B). The diagonals of the resulting confusion matrices (representing the Jaccard indices) were extracted and visualized using the R package ComplexHeatmap v2.10.0 (*Gu et al., 2016*).

## Analysis of chromatin states

The emission matrices from the ChromHMM models were read into R 4.1.2 (https://www.R-project.org/.) and plotted using the R packages ggplot2 from tidyverse (doi:10.21105/joss.01686) and package ComplexHeatmap v2.10.0 (*Gu et al., 2016*). The state assignment files from ChromHMM for the different models and tissues/genotypes were read in and analyzed using packages from tidyverse and valr v0.6.6 (*Riemondy et al., 2017*). To compare the wild-type and *ddm1* assignments from the concatenated model, separate files were joined. To this end, for each (200 bp) bin of the genome, the information about which state had been assigned in wild-type and in *ddm1*, respectively was combined. The state assignment datasets were then overlapped, using the function bed_intersect from valr, with regions for genomic features defined in the file `Araport11_GFF3_genes_trans-posons.Jun2016.gff` downloaded from (https://www.arabidopsis.org/download_files/Genes/Araport11_genome_release/archived/Araport11_GFF3_genes_transposons.Jun2016.gff.gz). The transcriptome, methylome, and DNase I datasets were integrated to generate plots using R and the ggplot2 package from tidyverse. All box plots show the data excluding outliers.

## Comparison of states from extensive and concatenated models

To compare the states from the two tissue models with the chromatin types defined in the extensive model, the states assigned to seedlings by the two tissue model were overlapped with the chromatin types from the extensive model. For the comparison of the wild-type and *ddm1* mutant model states with the chromatin types from the extensive model, the states assigned to the wild-type was used. The genomic overlaps were calculated using bed_intersect from valr v0.6.6 (*Riemondy et al., 2017*). States overlapping one chromatin type with >66% and all others with <20% were assigned the largest overlapping chromatin type. In other cases, the state was classified as mixed (*Figure 3—figure supplement 1B*; *Figure 3—figure supplement 1C*).

## Analysis of chromatin state changes in *ddm1* mutant

The Jaccard index for each state was calculated as the total length of regions assigned to that state in both wild-type and mutant divided by the combined length of all regions belonging to that state in at least one of the two genotypes. The overlap was calculated as the total length of regions assigned to that state in both wild-type and mutant divided by the total length of regions assigned to that state in wild-type. The size fold change was calculated as the total length of regions assigned to that state in *ddm1* mutant divided by the total length of all regions assigned to the state in the wild-type.

Or, formally, if $B_{w,m}$ is the total number of bins that are assigned to state *w* in the wild-type and state *m* in *ddm1*, then the (JI) Jaccard index, (O) overlap, and (FC) size change for states can be calculated as:

$$JI_s = \frac{B_{s,s}}{B_{s,.} + B_{.,s} - B_{s,s}}, \ O_s = \frac{B_{s,s}}{B_{s,.}}, \ FC_s = log_2\left(\frac{B_{.,s}}{B_{s,.}}\right)$$

The plots were generated using R and the ggplot2 package from tidyverse. The metaplots were generated using Deeptools v3.1.2 (*Ramírez et al., 2016*). Using the bamCompare function of Deeptools v3.1.2 bigWig files were generated by normalizing ChIP samples with input/H3. These bigWig files were used for plotting the metaplot heatmap (*Figure 4A*) and profile plots (*Figure 4—figure supplement 1A and B*) using deeptools v3.1.2 plotHeatmap and plotProfile functions, respectively.

### RNA-seq of WT seedlings

Total RNA was extracted with Spectrum plant total RNA kit (Sigma Aldrich) from 10 day seedlings of WT Col-0. A DNA-free DNA removal kit was used to remove contaminated DNA (Thermo Fisher Scientific). RNA-seq poly-A libraries were generated with NEBNext UltraII directional RNA library prep kit for Illumina (New England Biolabs) following the manufacturer's instructions. The libraries were sequenced with Illumina Hiseq 2500 to generate single-end 50 bp reads. Samples were prepared from three independent biological replicates. The RNA-seq data was processed as described in *Osakabe et al., 2021*.

### Methylation data

BS-seq data for WT Col-0 was downloaded from GSE39901 (*Stroud et al., 2013*). BS-seq reads were processed using the nf-core pipeline (*Ewels et al., 2023*) as described in *Pisupati et al., 2023*. Cutadapt v2.10 (*Martin, 2011*) was used to trim the adaptors (default parameters). Trimmed reads were then mapped to TAIR10 (Col-0) assembly using bismark v0.2.1 (*Krueger and Andrews, 2011*) allowing mismatches to 0.5. Methylation calls were performed using methylpy v1.3.7 (*Schultz et al., 2016*) on the aligned bam files. Methylation call bed files were used to calculate average methylation over chromatin state bed regions.

### DNase I-seq

We downloaded processed DNaseI - seq bigwig files data from GEO series GSE53322 for WT Col-0 (GSM1289358) (*Sullivan et al., 2014*). Bedtools map v2.3 (*Quinlan, 2014*) was used to calculate the averaged signal over bed regions in each chromatin state. Box plots were generated in R v3.5 using ggplot2 to compare the accessibility across chromatin states.

### RNA-seq data analysis of wild-type and *ddm1* mutant leaves

We used the wild-type and *ddm1* mutant RNA-seq data from GSE150435. The data was processed as described in *Osakabe et al., 2021*. Before grouping the TE genes into five groups we first excluded all TE genes showing any expression in wild-type (tpm >0). TE genes without expression (tpm <0.1) in *ddm1* mutants were put in the 'no expression' group. The remaining TE genes were divided into four quartiles based on the tpm values in the *ddm1* mutant.

## Acknowledgements

FB acknowledges support from the next-generation sequencing and PlantS facilities at the Vienna BioCenter Core Facilities (VBCF), the BioOptics facility, the Proteomics facility, and Molecular Biology Services from the Institute for Molecular Pathology (IMP), the Molecular Biology Services at the GMI. We thank all members of the Berger Laboratory for their technical help. We thank J Matthew Watson for insightful discussions and critical reading of the manuscript. This work was supported by the Japan Society for the Promotion of Science (JSPS) Overseas Research Fellowships [to AO], the Austrian Science Fund (FWF): M2539-B21 [to AO], P32054, P30802, and P33380 [to FB], and DK1238 [to BJ], the Austrian Academy of Sciences [to FB, ZJL, EA., SA, RP and RY], JSPS Grant-in-Aid for Research Activity Start-up (JP21K20628), JSPS Grant-in-Aid for Transformative Research Areas (JP22H05172 and JP22H05178), and the Japan Science and Technology Agency (JST) PRESTO (JPMJPR20K3) [to AO].

# Additional information

## Funding

| Funder | Grant reference number | Author |
|---|---|---|
| Austrian Science Fund | M2539-B21 | Akihisa Osakabe |
| Austrian Science Fund | P32054 | Frédéric Berger |
| Austrian Science Fund | P30802 | Frédéric Berger |
| Austrian Science Fund | P33380 | Frédéric Berger |
| Austrian Science Fund | DK1238 | Bhagyshree Jamge |
| Japan Society for the Promotion of Science | JP21K20628 | Akihisa Osakabe |
| Japan Society for the Promotion of Science | JP22H05172 | Akihisa Osakabe |
| Japan Society for the Promotion of Science | JP22H05178 | Akihisa Osakabe |
| Japan Science and Technology Agency | JPMJPR20K3 | Akihisa Osakabe |
| Austrian Academy of Sciences | | Frédéric Berger<br>Zdravko J Lorković<br>Elin Axelsson<br>Svetlana Akimcheva<br>Ramesh Yelagandula |
| Japan Society for the Promotion of Science | Overseas Research Fellowships | Akihisa Osakabe |

The funders had no role in study design, data collection and interpretation, or the decision to submit the work for publication.

## Author contributions

Bhagyshree Jamge, Formal analysis, Validation, Investigation, Visualization, Methodology, Writing - review and editing; Zdravko J Lorković, Data curation, Formal analysis, Validation, Investigation, Visualization, Methodology, Writing - original draft, Writing - review and editing; Elin Axelsson, Resources, Data curation, Software, Formal analysis, Supervision, Validation, Investigation, Visualization, Methodology, Project administration, Writing - review and editing; Akihisa Osakabe, Data curation, Formal analysis, Validation, Investigation, Visualization, Methodology, Writing - review and editing; Vikas Shukla, Resources, Data curation, Software, Formal analysis; Ramesh Yelagandula, Annika Luisa Kuehn, Investigation; Svetlana Akimcheva, Resources, Methodology, Project administration; Frédéric Berger, Conceptualization, Supervision, Funding acquisition, Writing - original draft, Project administration, Writing - review and editing

## Author ORCIDs

Elin Axelsson http://orcid.org/0000-0003-4382-1880
Vikas Shukla http://orcid.org/0000-0001-8739-8610
Ramesh Yelagandula http://orcid.org/0000-0001-7418-4322
Frédéric Berger http://orcid.org/0000-0002-3609-8260

Reviewer #1 (Public Review): https://doi.org/10.7554/eLife.87714.3.sa1
Reviewer #2 (Public Review): https://doi.org/10.7554/eLife.87714.3.sa2
Reviewer #3 (Public Review): https://doi.org/10.7554/eLife.87714.3.sa3
Reviewer #4 (Public Review): https://doi.org/10.7554/eLife.87714.3.sa4
Author Response https://doi.org/10.7554/eLife.87714.3.sa5

## Additional files

### Supplementary files
• Supplementary file 1. Mapping statistics. The file includes output statistics for mapping and trimming of all data included in the manuscript.
• MDAR checklist

### Data availability

The genome-wide sequencing (ChIP-seq) generated for this study as well as published datasets (ChIP-seq, RNA-seq) utilized to support the findings in this study have been deposited on NCBI's Gene Expression Omnibus (GEO) with the accession number of subseries GSE226469, GSE231398, GSE150434 and GSE150433. Super series associated with this study and published data are GSE231408 and GSE150436. Source data for all the main figures as well as supplementary figures have been provided with this manuscript. All the code used to analyze the genome-wide sequencing data presented in this study, as described in the Methods are available at https://github.com/Gregor-Mendel-Institute/jamge_states_2023 (copy archived at swh:1:rev:5254163528f971a97c4c514daca96f01b11e545a).

The following datasets were generated:

| Author(s) | Year | Dataset title | Dataset URL | Database and Identifier |
|---|---|---|---|---|
| Jamge B | 2023 | Chromatin immunoprecipitation DNA-sequencing (ChIP-seq) for histone H2A variants and histone modifications in WT Col-0 seedlings | https://www.ncbi.nlm.nih.gov/geo/query/acc.cgi?acc=GSE226469 | NCBI Gene Expression Ominbus, GSE226469 |
| Jamge B | 2023 | ChIP-seq for Histone variants and Histone modifictaions in WT and ddm1(-/-) in 1st generation Leaves | https://www.ncbi.nlm.nih.gov/geo/query/acc.cgi?acc=GSE231398 | NCBI Gene Expression Omnibus, GSE231398 |
| Jamge B | 2023 | ChIP-seq for Histone variants and Histone modifications | https://www.ncbi.nlm.nih.gov/geo/query/acc.cgi?acc=GSE231408 | NCBI Gene Expression Omnibus, GSE231408 |

The following previously published datasets were used:

| Author(s) | Year | Dataset title | Dataset URL | Database and Identifier |
|---|---|---|---|---|
| Jamge B | 2021 | The chromatin remodeler DDM1 silences transposons through deposition of the histone variant H2A.W | https://www.ncbi.nlm.nih.gov/geo/query/acc.cgi?acc=GSE150436 | NCBI Gene Expression Omnibus, GSE150436 |
| Jamge B, Osakabe A | 2021 | RNA-seq for WT and ddm1(-/-) to evaluate transposon expression in ddm1 1st generation | https://www.ncbi.nlm.nih.gov/geo/query/acc.cgi?acc=GSE150433 | NCBI Gene Expression Omnibus, GSE150433 |
| Jamge B, Osakabe A | 2021 | ChIP-seq for Histone Variant H2A.W, H2A.X, H1 and H3K9me2 in WT and ddm1(-/-) to quantify loss of H2A.W in ddm1 1st generation | https://www.ncbi.nlm.nih.gov/geo/query/acc.cgi?acc=GSE150434 | NCBI Gene Expression Omnibus, GSE150434 |

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
