## [Editor Report · eLife assessment]

This study presents an **important** description on the dynamics of histone variant exchange controlling the organization of the chromatin state of the Arabidopsis genome, combining the analysis of histone variants, histone modification, and chromatin states. The evidence supporting the claims of the authors is **compelling**. This work will be of great interest to those in the field of epigenetics and chromatin biology.

---

## [Referee Report · Reviewer #1 (Public Review)]

Jamge et al. sought to identify the relationships between histone variants and histone modifications in Arabidopsis by systematic genomic profiling of 13 histone variants and 12 histone modifications to define a set of "chromatin states". They find that H2A variants are key factors defining the major chromatin types (euchromatin, facultative heterochromatin, and constitutive heterochromatin) and that loss of the DDM1 chromatin remodeler leads to loss of typical constitutive heterochromatin and replacement of this state with features common to genes in euchromatin and facultative heterochromatin. This study deepens our understanding of how histone variants shape the Arabidopsis epigenome and provides a wealth of data for other researchers to explore.

Strengths:

1. The manuscript provides convincing evidence supporting the claims that: (A) Arabidopsis nucleosomes are homotypic for H2A variants and heterotypic for H3 variants, (B) that H3 variants are not associated with specific H2A variants, and (C) H2A variants are strongly associated with specific histone post-translational modifications (PTMs) while H3 variants show no such strong associations with specific PTMs. These are important findings that contrast with previous observations in animal systems and suggest differences in plant and animal chromatin dynamics.

2. The authors also performed comprehensive epigenomic profiling of all H2A, H2B, and H3 variants and 12 histone PTMs to produce a Hidden Markov Model-based chromatin state map. These studies revealed that histone H2A variants are as important as histone PTMs in defining the various chromatin states, which is unexpected and of high significance.

3. The authors show that in ddm1 mutants, normally heterochromatic transposable element (TE) genes lose H2A.W and gain H2A.Z, along with the facultative heterochromatin and euchromatin signatures associated with H2A.Z at silent and expressed genes, respectively.

---

## [Referee Report · Reviewer #2 (Public Review)]

Jamge et al. set out to delineate the relationship between histone variants, histone modifications and chromatin states in Arabidopsis seedlings and leaves. A strength of the study is its use of multiple types of data: the authors present mass-spec, immunoblotting and ChIP-seq from histone variants and histone modifications. They confirm the association between certain marks and variants, in particular for H2A, and nicely describe the loss of constitutive heterochromatin in the ddm1 mutant.

Overall, this study nicely illustrates that, in Arabidopsis, histone variants (and H2A variants in particular) display specificity in modifications and genomic locations, and correlate with some chromatin sub-states. This encourages future work in epigenomics to consider histone variants with as much attention as histone modifications.

---

## [Referee Report · Reviewer #3 (Public Review)]

How chromatin state is defined is an important question in the epigenetics field. Here, Jamge et al. proposed that the dynamics of histone variant exchange control the organization of histone modifications into chromatin states. They found (1) there is a tight association between H2A variants and histone modifications; (2) H2A variants are major factors that differentiate euchromatin, facultative heterochromatin, and constitutive heterochromatin; (3) the mutation in DDM1, a remodeler of H2A variants, causes the mis-assembly of chromatin states in TE region. The topic of this paper is of general interest and the results are novel.

Overall, the paper is well-written and the results are clearly presented. The biochemical analysis part is solid.

---

## [Referee Report · Reviewer #4 (Public Review)]

This work aims at analyzing the impact of histone variants and histone modifications on chromatin states of the Arabidopsis genome. Authors claim that histone variants are as significant as histone modifications in determining chromatin states. They also study the effect of mutations in the DDM1 gene on the exchange of H2A.Z to H2A.W, which convert the silent state of transposons into a chromatin state normally found on protein coding genes.

This is an interesting and well done study on the organization of the Arabidopsis genome in different chromatin states, adding to the previous reports on this issue.

---

## [Author Response]

The following is the authors’ response to the original reviews.

**Reviewer #1 (Public Review):**
Jamge et al. sought to identify the relationships between histone variants and histone modifications in Arabidopsis by systematic genomic profiling of 13 histone variants and 12 histone modifications to define a set of "chromatin states". They find that H2A variants are key factors defining the major chromatin types (euchromatin, facultative heterochromatin, and constitutive heterochromatin) and that loss of the DDM1 chromatin remodeler leads to loss of typical constitutive heterochromatin and replacement of this state with features common to genes in euchromatin and facultative heterochromatin. This study deepens our understanding of how histone variants shape the Arabidopsis epigenome and provides a wealth of data for other researchers to explore.Strengths:1. The manuscript provides convincing evidence supporting the claims that: (A) Arabidopsis nucleosomes are homotypic for H2A variants and heterotypic for H3 variants, (B) that H3 variants are not associated with specific H2A variants, and (C) H2A variants are strongly associated with specific histone post-translational modifications (PTMs) while H3 variants show no such strong associations with specific PTMs. These are important findings that contrast with previous observations in animal systems and suggest differences in plant and animal chromatin dynamics.2. The authors also performed comprehensive epigenomic profiling of all H2A, H2B, and H3 variants and 12 histone PTMs to produce a Hidden Markov Model-based chromatin state map. These studies revealed that histone H2A variants are as important as histone PTMs in defining the various chromatin states, which is unexpected and of high significance.3. The authors show that in ddm1 mutants, normally heterochromatic transposable element (TE) genes lose H2A.W and gain H2A.Z, along with the facultative heterochromatin and euchromatin signatures associated with H2A.Z at silent and expressed genes, respectively.Weaknesses:1. Following up on the finding that H2A.Z replaces H2A.W at TE genes in ddm1 mutants, the authors provide in vitro evidence that DDM1 binds to H2A.Z-H2B dimers. These results are taken together to conclude that DDM1 normally removes H2A.Z-H2B dimers from nucleosomes at TE genes and replaces them with H2A.W-H2B dimers. However, the evidence for this model is circumstantial and such a model raises a variety of other questions that are not addressed by the authors.

The Reviewer raises a series of interesting questions. We proposed that DDM1 exchanges H2A.Z to H2A.W because it is the simplest model and also because LSH - the mammalian ortholog of DDM1 exchanges H2A to macroH2A. However we do stress in the revised manuscript that this is a model and other possible models that could involve chaperones and additional remodelers are possible. Addressing why the loss of DDM1 results in a net exchange of H2A.W to H2A.Z is not the purpose of this study. Here we use the perturbation caused by ddm1 as a means to address the importance of the dynamics exchange of H2A variants in setting up the chromatin states. We do observe that perturbing this dynamic exchange causes an important perturbation of chromatin states. This further supports our main conclusion: H2A variants dynamics are one important factor that organizes chromatin states.

For example: if DDM1 does remove H2A.Z from TE genes, how does H2A.Z normally come to occupy these sites, given that they are highly DNA methylated and that H2A.Z is known to anticorrelate with DNA methylation in plants and animals?

The anticorrelation between H2A.Z and DNA methylation is observed at steady state. The exchange of H2A.Z to H2A.W that results from the action of DDM1 would indeed remove unwanted H2A.Z from regions occupied by DNA methylation as suggested by the Reviewer.

Given that H2A.Z does not accumulate in TEs in h2a.w mutants, how would H2A.X and H2A instead become enriched at these sites if DDM1 cannot bind these forms of H2A?

This is a valid question: We envisage that H2A.X and H2A are deposited by remodelers and chaperones other than DDM1 in the h2a.w mutant.

Given that there are no apparent regions with common sequence between H2A.Z and H2A.W variants that are not also shared with other H2A classes, how would DDM1 selectively bind to H2A.W-H2B and H2A.Z-H2B dimers to the exclusion of H2A(.X)-H2B dimers?

It was shown by the Muegge Lab both in vitro and in vivo that LSH - the mammalian ortholog of DDM1 binds to macroH2A and H2A, and these two H2A variants do not share similar specific region. Yet it remains to determine which region of H2A.Z and H2A.W binds to DDM1, which does not fit in the scope of this study.

**Reviewer #2 (Public Review):**
Jamge et al. set out to delineate the relationship between histone variants, histone modifications and chromatin states in Arabidopsis seedlings and leaves. A strength of the study is its use of multiple types of data: the authors present mass-spec, immunoblotting and ChIPseq from histone variants and histone modifications. They confirm the association between certain marks and variants, in particular for H2A, and nicely describe the loss of constitutive heterochromatin in the ddm1 mutant.The support for some of the conclusions is weak. The title of the discussion, "histone variants drive the overall organization of chromatin states" implies a causation which wasn't investigated, and overstates the finding that some broad chromatin states can be further subdivided when one considers histone variants (adding variables to the model).

We have removed subtitles in the discussion and have taken care to avoid over simplified statements.

Adding variables to a ChromHMM model naturally increases the complexity of the models that can be built, however it is difficult to objectively define which level of complexity is optimal. The differences between states may be subtle to the point that they may be considered redundant. The authors claim that the sub-states they define are biologically important, but provide little evidence to support this claim. It is not obvious whether the 26 states model is much more useful than a 9-states model. Removing variables naturally affects the definition of states that depend on these variables, but it is also hard to define the biological significance of that change. This sensitivity analysis is thus not very developed.

We agree that adding more input tracks/ data will increase the complexity.

But we would like to mention the differences of this study and the 9-state model,

1. We have included the histone variants which have been previously missed in chromatin state definition.

2. The previous 9-state model used data from different tissue types. In this study all the data generated and analyzed is from seedlings.

3. Increasing the number of states allowed us to resolve heterochromatin states compared to 9-state model which was previously missed. (BioRXiv)

4. The biological relevance of the 26 states model is analyzed and described in depth (States BioRxiv paper).

In addition we have now updated the Figure 2F to include a more direct comparison of marks used in both models. And we have expanded the description in the methods section and our reasoning behind using 26 state model to be analyzed in depth.

There are issues with the logical sequence of arguments in Fig1 and Fig3. Fig1A shows that nucleosomes often contain both H3.1 and H3.3. Therefore pulling-down H3.1-containing nucleosomes also pulls down H3.3 and whether specific H2A variants associated with H3.1 cannot be answered in this way (Fig1B).

We thank the Reviewer for point this out. If 60% of nucleosomes are homotypic and if they would associate with a specific H2A variant this would be clearly visible on WB as a much stronger band. Also, the MS data presented in Figure1 figure supplement 1D clearly show that all H2A variants associate with both H3.1 and H3.3. We have included in the revised version more detailed explanation to clarify this point.

The same issue likely carries to the investigation of the association with H3 modifications if Fig1C and 1D, since the H3.1-HA pull-down also pulls down endogenous H3.1 (so presumably the rest of the nucleosome, with H3.3, as well).

We disagree on this point. The H3 band corresponding to the transgene copy is eitherH3.1 or H3.3, so all signals on upper band (T) in Figure 1C are associated with either H3.1 (H3.1 IP) or H3.3 (H3.3 IP), thus unambiguously showing that all modifications we analyzed are present on both H3.1 and H3.3. Furthermore, data shown in Figure 1D and E, where we analyzed modifications on K27 and K36 which are in the H3 region that can be distinguished between H3.1 and H3.3 by MS clearly demonstrate that these modifications are present on both H3.1 and H3.3. In order to make this clearer, we also extended the description of this part in the Results section to emphasize this.

In Fig3, the conclusion that it is the loss of H2A.Z -> H2A.W exchange in the ddm1 mutant that causes loss of constitutive heterochromatin is rushed. The fact that the h2a.w mutant does not recapitulate the loss of constitutive heterochromatin seen in ddm1 argues against this interpretation.

We agree that at first the minimal impact of the loss of H2A.W alone is surprising. However, we point to the preprint https://www.biorxiv.org/content/10.1101/2022.05.31.493688v1. There it is shown that the joint loss of H2A.W and H3K9 methylation (also observed in ddm1) affects silencing of a large range of transposons that also lose silencing in ddm1.

It's also difficult to conclude about the importance of dynamic exchanges when the ddm1 mutation has been present for generations and the chromatin landscape has fully readapted. Further work is needed to support the authors' hypothesis.

We apologize that the Reviewer could not find the information regarding the origin of ddm1 mutant material. We did not use a mutant where ddm1 mutations was kept for generations. We were in fact very careful on this point and used leaves from ddm1 first homozygous plants segregated from heterozygous ddm1 kept heterozygous.

The study also relies on a large number of custom (polyclonal) antibodies with no public validation data. Lack of specificity, a common issue with antibodies, would muddle the interpretation of the data.

We added information about validation of custom made antibodies into Methods:”Specificities of custom made polyclonal antibodies against Arabidopsis H2A.Z.9, H2A.X, H2A.W.6, H2A.13, H2A.W.7, H2Bs, and linker histone H1 were validated in previouspublications (Yelagandula et al., 2014; Lorkovic et al., 2017; Jiang et al., 2020; Osakabe et al., 2021).“ For H2A.2 and H2A.Z.11 antibodies we provide validation data as Figure 2 figure supplement 1.

Overall, this study nicely illustrates that, in Arabidopsis, histone variants (and H2A variants in particular) display specificity in modifications and genomic locations, and correlate with some chromatin sub-states. This encourages future work in epigenomics to consider histone variants with as much attention as histone modifications.
**Reviewer #3 (Public Review):**
How chromatin state is defined is an important question in the epigenetics field. Here, Jamge et al. proposed that the dynamics of histone variant exchange control the organization of histone modifications into chromatin states. They found (1) there is a tight association between H2A variants and histone modifications; (2) H2A variants are major factors that differentiate euchromatin, facultative heterochromatin, and constitutive heterochromatin; (3) the mutation in DDM1, a remodeler of H2A variants, causes the mis-assembly of chromatin states in TE region. The topic of this paper is of general interest and results are novel.Overall, the paper is well-written and results are clearly presented. The biochemical analysis part is solid.
**Reviewer #4 (Public Review):**
This work aims at analyzing the impact of histone variants and histone modifications on chromatin states of the Arabidopsis genome. Authors claim that histone variants are as significant as histone modifications in determining chromatin states. They also study the effect of mutations in the DDM1 gene on the exchange of H2A.Z to H2A.W, which convert the silent state of transposons into a chromatin state normally found on protein coding genes.This is an interesting and well done study on the organization of the Arabidopsis genome in different chromatin states, adding to the previous reports on this issue.
**Reviewer #1 (Recommendations For The Authors):**
1. The rationale for switching from using 10-day old seedlings for chromatin profiling to using mature leaves in Figure 3 and beyond is not explained and introduces additional complexity into the analyses. The reasoning should be clearly explained in the text, and there are several additional suggestions or questions related to this that should be addressed:

This was done for practical reasons. We had already obtained some profiles of marks in ddm1 mutants and extended the dataset using the same stage of development because this tied this study with our previous study. Using different stages of development provides an additional benefit. The same chromatin states are observed in 10 day old seedlings and leaves of older plants. Constitutive heterochromatin is occupied by the same chromatin states and logically euchromatin is positioned on different genes as expected by the distinct pattern of gene expression at the two stages of development.

A) In the 16-state model (Figure 3A), euchromatin states were not well defined compared to the 26-state model. Why did the authors not profile these marks also, and could this explain why ddm1 mutants did not show a significant effect on euchromatin states in this model?

We apologize for the lack of detailed explanation: In our previous study we used leaves of five weeks ld plants to show the impact of ddm1 on the profiles of H2A.W.6, H2A.X, H1, H3K9me2, H3K36me3 and H3K27me3 in leaves (Jamge, Osakabe et al., 2021). This study showed that DDM1 causes the deposition of H2A.W.6 to heterochromatin and we thus used leaves to extend this investigation to the two other marks of heterochromatin (constitutive or facultative) H3K9me1, H2A.W.7 and H2A.Z.9 and H2A.Z.11.

B) The authors state that the tissue types do not impact the definition of chromatin states. However, there is a clear difference in the portion of the genome occupied by each chromatin state between leaf and seedling (states 1, 5, 8, 13, and 14; Figure S3A).

We had missed a comment on supFig3B and have now provided more explanation: “Although the composition of the chromatin states did not vary significantly between seedlings and leaves, each state occupied a similar proportion of the genome in seedling or leaves to the exception of state 5 present primarily in leaves and state 13 only present in seedlings (Figure 3 figure supplement 3A, right column with green bars) and the euchromatin states occupied different genes (Figure 3 figure supplement 3B) as expected by the dissimilar transcriptomes of these two developmental stages.”

2. The naming of supplemental figures throughout the text is confusing as the legends refer to them as "Figure SX" but they are called out in the text as "Figure X figure supplement XA-B".The eLifeconvention is "Figure X figure supplement XA-B".

This was changed.

3. In Figure 4, Panel D is mislabeled as C in the figure, and C is lacking a label.4. Please remove the word "the" from the title.

This was done

**Reviewer #2 (Recommendations For The Authors):**
Fig1D legend should also mention K37.

This was corrected.

Fig2F legend should say "no H3 modifications" rather than "no histone modifications"This was corrected.Fig4 labels C/D do not correspond to the legend. D is missing and C should go to the ddm1 stacked barplot.

This was corrected.

H3 variants analysis: Taking the relative abundance of H3.1 and H3.3 (and transgenes) into account would be useful to interpret the results of the nucleosome composition results. If they are at equivalent amounts, the null hypothesis of independent association would give 50% heterotypic nucleosomes and 50% homotypic.

This is a valid comment. In an ideal system the last statement would be correct, but this does not take into account chromatin dynamics associated with replication, transcription, etc. Also, total amounts of H3.1 and H3.3 in tissue we used for the experiment is not known. It could possibly be inferred from RNAseq data, but if this would reflect real amounts of the protein is highly questionable. In Arabidopsis there are 5 H3.1 genes and 3 H3.3 genes. Nevertheless, we recalculated data for H3.1 and H3.3 and this has been updated in the main text (~60% of H3.1 and ~42% of H3.3 immunoprecipitated nucleosomes contained both H3 variants). Thus, from the available data these numbers are the best we can get.

p. 5 bottom paragraph. Repetition.

This was corrected

p12. The reference to LSH is dropped in without making clear how it is relevant. Expand on mechanism to suggest similar DDM1 mechanism?

This section was expanded to provide more background in the interpretation of the results.

p13. inversion between H2A.W and H2A.Z in "the loss of DDM1 prevents the replacement of H2A.W by H2A.Z".

This was corrected

p13. make it clear that the last sentence of the results is a working model, not a fully backed up conclusion.

Alternative models are mentioned in this section and in the discussion in the revised version.

p14 middle paragraph. Not clear what "in silico simulation" refers to. Simply chromatin-state classification with ChromHMM?

This refers to the Jacard index calculation in Fig. 2F that models the impact of the loss of H2A variants (or other elements of chromatin) on the definition of chromatin states by ChromHMM. This is now clarified.

p14 bottom paragraph: the H2A.Z tail repression of ubiquitin ligase but its being the favoured substrate for H2AK121Ub is apparently contradictory. Can this be explained?

This refers to H2B Ubiquitination and is now clarified

p15. Correlation between variants and modifications/chromatin states does not necessarily mean causation.

We agree and have improved the revised version in this respect.

p15 "forward feedback loop" is ambiguous (is it a feed-forward loop? A feedback loop?), just use "positive feedback loop".

This was corrected.

p23 top "$(Ingouff et al)" doesn't seem properly formatted.

This reference did not belong there and has been removed.

Data availability: GSE226469 is not public. The manuscript also mentions availability of source data for all the main figures, but I could not find it. It would be great to make the code publicly available too.

All the data and code will be public upon posting the revised version of the manuscript.

**Reviewer #3 (Recommendations For The Authors):**
My major concern is authors only used DDM1 as an example to show that the exchange of the histone variant contributes to definition and distribution of chromatin state on transposons (i.e., constitutive heterochromatin regions associated with H2A.W). Readers may wonder whether similar mechanisms also work at the euchromatin region. This point should be clearly discussed and mentioned in the Results (for example, cite recent work on INO80).

We discuss the impact of other remodelers in the Discussion in the revised version. We hope that the reviewer will understand that doing a study on the impact of other remodelers on chromatin states which would require dozens of new ChIP profiles and is clearly beyond the scope of revising a manuscript.

Minor:1. Fig. 2A and 2B, what does color mean? I guess the color code is referred to chromatin states (Fig. 2F).

We have clarified on Figure 2A the attribution of a specific color to each chromatin state. This same color is used also in other panels of Figures 2 and S2.

2. Supplemental Figures: All the figure panels should be on the same page.

We rearranged supplemental figures so that each figure fits on one page. In places where this was not possible, we created additional supplemental figures.

3. "We observed that increasing state numbers from 26 to 27 gave rise to biologically redundant states.": Where are the data? Fig S2A? This figure is hard to understand.

In the updated manuscript, we have described the legend and the methods for FigS2A in more detail.

**Reviewer #4 (Recommendations For The Authors):**
A general concern refers to the text that frequently falls into excessive oversimplifications and/or overstatements, with the danger of being misleading for the reader. This needs to be thoroughly revised.

We added more careful statements and proposed alternative models when it was possible.

Specific comments.1. Fig 1A. Authors found the ~40% of nucleosomes contained both H3.1 and H3.3. This is a significant finding that deserves a more detailed comment.

We now provide a more detailed description of IP and MS data presented in Figure 1. This should also help to avoid oversimplifications and/or overstatements as criticized in a general comment.

2. Fig 1C. "H3. And H3.3 bore the same sets and comparable levels of methylation and acetylation...". Too general statement, please specify. Is this also the case for H3K9me2? Others?

We did describe this part into more detail to emphasize more precisely what Figure 1 shows. We also included data on K9me into Figure 1 figure supplement 1H.

3. Fig 1D. Could you confirm the high level of H3K27me1 on H3.3?

H3K27me1 data are shown both by WB (Figure 1C) and Mass spectrometry (Figure 1D and E). We also provide a possible explanation for high levels of this mark on H3.3 by taking into account the fact that H3K27me1 is also produced by demethylation of H3K27me3 by JMJ demethylases.

4. All WB in Fig 1. They need to be quantified and normalized (plus statistical analysis) in order to provide strong support to the conclusions.

The conclusion of all WB are supported by quantified Mass spectrometry data and many WB were even repeatedly shown in Figure 1F (for example IPs for H2A variants and a large set of H3 marks used for WBs) with the same results. Also, association of H3K4me3 and H3K36me3 with H2A variants was analyzed in both ways (Figure 1F); IPs of variants and WBs of variants and marks and IPs of marks and WBs of marks and variants. For most of the data we do not have more than two repeats, so statistical analysis may not be possible.

Nevertheless, we are convinced that our major conclusions from data presented in Figure 1 andSupporting figure 1 (these are: that H3 variants form both homotypic and heterotypic nucleosomes, that H3 marks do not preferentially associate with H3 variants but some of them do so with H2A variants and that H3 modifications show very complex pattern of associations with each other) are fully valid as they were drawn from two orthogonal approaches and further supported by the chromatin states identified.

5. Fig. 2A. Authors focus on "the most parsimonious model" based on 26 chromatin states. This needs to be justified in a more explicit manner. It is surprising that this number emerges for an analysis of 27 independent variants and marks. What are the differences in the conclusions when other number of states are used? See also below (reduced number of number derived from the "concatenated model").

Why 26 states were chosen is now explained in great details in the method section. Since to the exception of H2A variants that are invariably homotypic, nucleosomes can be heterotypic for all other histone variants and histone modifications, the random combination of the 27 marks in one nucleosome representing one states is 4 H2A (without the subtypes) x 4H3 x 2H1 x 2(power16) (for each mark) which is well above the circa 26 states observed. This shows that our probabilistic model reduces the potential complexity of a theorical random association in a remarkable manner.

6. As a summary, it would be very helpful to generated a table (or similar) where is proposed chromatin state is ascribed to functional genomic elements.

This aspect of the work is presented in a preprint where the biological association with the chromatin is described in details. See Jamge et al 2002, https://www.biorxiv.org/content/10.1101/2022.06.02.494419v1.

7. Fig 2F (and S2B). A comprehensive comparison a various approaches should include others and estimate the Jaccard similarity index: (1) the same of marks and variants used in the Sequeira-Mendes et al paper, and (2) the subset of marks and variants added in this study. In this way, a direct evaluation of the contributions could be more properly made.

We thank the reviewer for this suggestion and have now included a new column with the combination of marks and variants as used in Sequeira-Mendes et al., 2014 (see Figure 2F). These data clearly demonstrate that adding histone variants significantly contribute to the definition of chromatin states.

8. Fig. 3. Explain in more detail the concatenated model used here. Does the reduction in the number of chromatin states mean that the other do not add new information?

ChromHMM concatenated model allows to identify common definition of chromatin state in multiple tissue types. Here multiple cell types are concatenated leading to a shared definition of chromatin states, but specific to each cell type.

In our paper we used the concatenated model to identify common chromatin states in two different genotypes (WT and ddm1). The data for WT and ddm1 was obtained from leaves. As we had a limited number of ChIP-seq profiles in the leaves dataset The complexity of the concatenated model was also reduced compared to the extensive 26 chromatin state model. We chose to analyze 16-states in the concatenated model because this was the minimal number of states that gave rise to a similar complexity of heterochromatic states.

9. The ddm1 mutant. The text in page 14 is a bit confusing. It seems that H2A.Z is deposited on TEs and the exchanged by the H2A.W.

We have provided additional alternative models that could explain our observations.

10. Page 15: link between H2A.Z and H3K27me3. Gomez-Zambrano et al 2018, cited in the text, found that only a relatively small subset of (putative) targets are common to H2A.Z and H3K27me3. How do authors reconcile this with their statement supporting a link between both of them?

We refer to Gomez-Zambranao et al to illustrate the link between H2A.Z and H2AK121ub so we do not understand this comment. The strong link between H2A.Z and H3K27me3 is shown without ambiguity by our work and also Carter et al., 2018.